# Intermediate Molecular Phenotypes to Identify Genetic Markers of Anthracycline-Induced Cardiotoxicity Risk

**DOI:** 10.3390/cells12151956

**Published:** 2023-07-27

**Authors:** Aurora Gómez-Vecino, Roberto Corchado-Cobos, Adrián Blanco-Gómez, Natalia García-Sancha, Sonia Castillo-Lluva, Ana Martín-García, Marina Mendiburu-Eliçabe, Carlos Prieto, Sara Ruiz-Pinto, Guillermo Pita, Alejandro Velasco-Ruiz, Carmen Patino-Alonso, Purificación Galindo-Villardón, María Linarejos Vera-Pedrosa, José Jalife, Jian-Hua Mao, Guillermo Macías de Plasencia, Andrés Castellanos-Martín, María del Mar Sáez-Freire, Susana Fraile-Martín, Telmo Rodrigues-Teixeira, Carmen García-Macías, Julie Milena Galvis-Jiménez, Asunción García-Sánchez, María Isidoro-García, Manuel Fuentes, María Begoña García-Cenador, Francisco Javier García-Criado, Juan Luis García-Hernández, María Ángeles Hernández-García, Juan Jesús Cruz-Hernández, César Augusto Rodríguez-Sánchez, Alejandro Martín García-Sancho, Estefanía Pérez-López, Antonio Pérez-Martínez, Federico Gutiérrez-Larraya, Antonio J. Cartón, José Ángel García-Sáenz, Ana Patiño-García, Miguel Martín, Teresa Alonso-Gordoa, Christof Vulsteke, Lieselot Croes, Sigrid Hatse, Thomas Van Brussel, Diether Lambrechts, Hans Wildiers, Hang Chang, Marina Holgado-Madruga, Anna González-Neira, Pedro L. Sánchez, Jesús Pérez Losada

**Affiliations:** 1Instituto de Biología Molecular y Celular del Cáncer (IBMCC-CIC), Universidad de Salamanca/CSIC, 37007 Salamanca, Spain; a.gomezvecino@usal.es (A.G.-V.); rober.corchado@usal.es (R.C.-C.); adrian.blancogomez@cruk.manchester.ac.uk (A.B.-G.); nataliagarciasancha@usal.es (N.G.-S.); marinamendiburu@usal.es (M.M.-E.); andres.castellanos@irbbarcelona.org (A.C.-M.); maria_del_mar@usal.es (M.d.M.S.-F.); jmgalvis@usal.es (J.M.G.-J.); mfuentes@usal.es (M.F.); jlgarcia@usal.es (J.L.G.-H.); jjcruz@usal.es (J.J.C.-H.); crodriguez@oncologiasalamanca.org (C.A.R.-S.); amartingar@usal.es (A.M.G.-S.); eperezl@saludcastillayleon.es (E.P.-L.); 2Instituto de Investigación Biosanitaria de Salamanca (IBSAL), 37007 Salamanca, Spain; anamartin.amg@gmail.com (A.M.-G.); carpatino@usal.es (C.P.-A.); pgalindo@usal.es (P.G.-V.); guillermohus@gmail.com (G.M.d.P.); chonela@usal.es (A.G.-S.); misidoro@usal.es (M.I.-G.); mbgc@usal.es (M.B.G.-C.); fjgc@usal.es (F.J.G.-C.); 3Departamento de Bioquímica y Biología Molecular, Facultad de Ciencias Químicas, Universidad Complutense, 28040 Madrid, Spain; sonica01@ucm.es; 4Instituto de Investigaciones Sanitarias San Carlos (IdISSC), 24040 Madrid, Spain; 5Servicio de Cardiología, Hospital Universitario de Salamanca, Universidad de Salamanca (CIBER.CV), 37007 Salamanca, Spain; 6Servicio de Bioinformática, Nucleus, Universidad de Salamanca, 37007 Salamanca, Spain; bioinformatica@usal.es; 7Human Genotyping Unit-CeGen, Human Cancer Genetics Programme, Spanish National Cancer Research Centre (CNIO), 28029 Madrid, Spain; sara.ruiz.pinto.86@gmail.com (S.R.-P.); gpita@cnio.es (G.P.); avelascor@cnio.es (A.V.-R.); 8Departamento de Estadística, Universidad de Salamanca, 37007 Salamanca, Spain; 9Escuela Superior Politécnica del Litoral, ESPOL, Centro de Estudios e Investigaciones Estadísticas, Campus Gustavo Galindo, Km. 30.5 Via Perimetral, Guayaquil P.O. Box 09-01-5863, Ecuador; 10Centro Nacional de Investigaciones Cardiovasculares (CNIC) Carlos III, 28029 Madrid, Spain; marialinajeros.vera@cnic.es (M.L.V.-P.); jose.jalife@cnic.es (J.J.); 11Biological Systems and Engineering Division, Lawrence Berkeley National Laboratory, Berkeley, CA 94720, USA; jhmao@lbl.gov; 12Berkeley Biomedical Data Science Center, Lawrence Berkeley National Laboratory, Berkeley, CA 92720, USA; 13Servicio de Patología Molecular Comparada, Instituto de Biología Molecular y Celular del Cáncer (IBMCC-CIC), Universidad de Salamanca, 37007 Salamanca, Spain; susannafm@usal.es (S.F.-M.); telmo.t@usal.es (T.R.-T.); janagm@usal.es (C.G.-M.); 14Instituto Nacional de Cancerología de Colombia, Bogotá 111511-110411001, Colombia; 15Servicio de Bioquímica Clínica, Hospital Universitario de Salamanca, 37007 Salamanca, Spain; 16Departamento de Medicina, Universidad de Salamanca, 37007 Salamanca, Spain; 17Unidad de Proteómica y Servicio General de Citometría de Flujo, Nucleus, Universidad de Salamanca, 37007 Salamanca, Spain; 18Departamento de Cirugía, Universidad de Salamanca, 37007 Salamanca, Spain; 19Servicio de Hematología, Hospital Universitario de Salamanca, CIBERONC, 37007 Salamanca, Spain; nines1812@hotmail.com; 20Servicio de Oncología, Hospital Universitario de Salamanca, 37007 Salamanca, Spain; 21Department of Paediatric Hemato-Oncology, Hospital Universitario La Paz, 28046 Madrid, Spain; antonioperezmartinez@yahoo.es; 22Department of Paediatric Cardiology, Hospital Universitario La Paz, 28046 Madrid, Spain; flarraya@yahoo.es (F.G.-L.); antoniocarton@yahoo.com (A.J.C.); 23Medical Oncology Service, Instituto de Investigación Sanitaria del Hospital Clínico San Carlos (IdISSC), Hospital Clínico San Carlos, 28040 Madrid, Spain; jagsaenz@yahoo.com; 24Department of Pediatrics, Solid Tumor Program, Centro de Investigación Médica Aplicada (CIMA), Universidad de Navarra, IdisNA, 31008 Pamplona, Spain; apatigar@unav.es; 25Department of Medicine, Gregorio Marañón Health Research Institute (IISGM), Centro de Investigación Biomédica en Red Oncológica (CIBERONC), Universidad Complutense, 28007 Madrid, Spain; mmartin@geicam.org; 26Department of Medical Oncology, Hospital Universitario Ramón y Cajal, 28034 Madrid, Spain; teressalonso25@gmail.com; 27Department of Molecular Imaging, Pathology, Radiotherapy and Oncology (MIPRO), Center for Oncological Research (CORE), Antwerp University, 2610 Antwerp, Belgium; christof.vulsteke@azmmsj.be (C.V.); lieselot.croes@azmmsj.be (L.C.); 28Department of Oncology, Integrated Cancer Center in Ghent, AZ Maria Middelares, 9000 Ghent, Belgium; 29Laboratory of Experimental Oncology (LEO), Department of Oncology, Department of General Medical Oncology, University Hospitals Leuven, Leuven Cancer Institute, Katholieke Universiteit (KU) Leuven, 3000 Leuven, Belgium; sigrid.hatse@kuleuven.be; 30VIB Center for Cancer Biology, VIB, 3000 Leuven, Belgium; thomas.vanbrussel@vib-kuleuven.be (T.V.B.); diether.lambrechts@vib-kuleuven.be (D.L.); 31Laboratory of Translational Genetics, Department of Human Genetics, Katholieke Universiteit (KU) Leuven, 3000 Leuven, Belgium; 32Department of General Medical Oncology and Multidisciplinary Breast Unit, Leuven Cancer Institute, and Laboratory of Experimental Oncology (LEO), Department of Oncology, Leuven Cancer Institute and University Hospital Leuven, Katholieke Universiteit (KU) Leuven, 3000 Leuven, Belgium; hans.wildiers@uzleuven.be; 33Departamento de Fisiología y Farmacología, Universidad de Salamanca, 37007 Salamanca, Spain; 34Instituto de Neurociencias de Castilla y León (INCyL), 37007 Salamanca, Spain

**Keywords:** anthracyclines, cardiotoxicity, complex genetic disease, intermediate molecular phenotypes, quantitative trait loci

## Abstract

Cardiotoxicity due to anthracyclines (CDA) affects cancer patients, but we cannot predict who may suffer from this complication. CDA is a complex trait with a polygenic component that is mainly unidentified. We propose that levels of intermediate molecular phenotypes (IMPs) in the myocardium associated with histopathological damage could explain CDA susceptibility, so variants of genes encoding these IMPs could identify patients susceptible to this complication. Thus, a genetically heterogeneous cohort of mice (*n* = 165) generated by backcrossing were treated with doxorubicin and docetaxel. We quantified heart fibrosis using an Ariol slide scanner and intramyocardial levels of IMPs using multiplex bead arrays and QPCR. We identified quantitative trait loci linked to IMPs (ipQTLs) and cdaQTLs via linkage analysis. In three cancer patient cohorts, CDA was quantified using echocardiography or Cardiac Magnetic Resonance. CDA behaves as a complex trait in the mouse cohort. IMP levels in the myocardium were associated with CDA. ipQTLs integrated into genetic models with cdaQTLs account for more CDA phenotypic variation than that explained by cda-QTLs alone. Allelic forms of genes encoding IMPs associated with CDA in mice, including AKT1, MAPK14, MAPK8, STAT3, CAS3, and TP53, are genetic determinants of CDA in patients. Two genetic risk scores for pediatric patients (*n* = 71) and women with breast cancer (*n* = 420) were generated using machine-learning Least Absolute Shrinkage and Selection Operator (LASSO) regression. Thus, IMPs associated with heart damage identify genetic markers of CDA risk, thereby allowing more personalized patient management.

## 1. Introduction

Cardiotoxicity due to anthracyclines (CDA) is a frequent problem in cancer patients that limits the efficacy of chemotherapy [1]. Long-term cardiotoxicity has repercussions for oncological disease prognosis [2] and a far-reaching impact on patient quality of life [3,4]. Anthracyclines produce acute necrosis and apoptosis of cardiomyocytes, leading to myocardial fibrosis, varying degrees of chronic functional damage, and even heart failure [5]. The degree of chronic CDA depends on many factors, including dose, age, gender, previous heart diseases, and combined treatment with other drugs [6,7]. Early cardiotoxicity detection and treatment are needed [8]. Since it is difficult to determine which patients will develop chronic CDA, efforts have been made to identify genetic risk markers. However, current evidence about the role of pharmacogenomic screening in anthracycline therapy is mixed because of the heterogeneity of the results obtained so far [9].

The diversity of the results observed when attempting to identify CDA genetic markers may result from the CDA being a complex polygenic disease or complex trait influenced by multiple genes contributing at systemic, tissue, cellular, and molecular levels. It is estimated that CDA has a proportion of phenotype variation due to genetics [10]. However, it is difficult to determine how best to measure the genetic influence of complex traits/diseases. The proportion of phenotypic variation of complex traits explained by the genetic component is known as narrow-sense heritability. The genetic variants associated with complex diseases account for only 10–20% of the phenotypic variation attributable to genetics. The genetic variants responsible for the remaining phenotypic variation cannot be identified and are considered missing heritability; identifying its origins remains a contentious matter [11]. Complex diseases and traits arise from multiple intermediate phenotypes or endophenotypes participating in their pathogenesis [12]. Also, intermediate phenotypes may themselves be complex traits. For instance, myocardial infarction is a complex-trait disease, susceptibility to which is determined using intermediate phenotypes such as arterial hypertension, hypercholesterolemia, and the response to tobacco exposure. However, these intermediate phenotypes are also complex traits influenced by lower-ranking intermediate phenotypes at the systemic, cellular, and molecular levels. Genetic determinants directly act on the intermediate molecular phenotypes implicit in this multidirectional network of intermediate phenotypes [12]. Indeed, a specific protein and RNA would be the simplest intermediate phenotypes, controlled by a few genes (including the coding gene, the genes encoding promoter-regulating transcription factors, and those controlling post-translational activity regulators, such as phosphorylation) [13]. The variable phenotypic presentation of complex diseases is related to the expressivity of their intermediate phenotypes [12,14]. Thus, different degrees of susceptibility to CDA could result from the variable expression of intermediate molecular phenotypes participating in CDA pathogenesis [15,16].

It was previously proposed that the missing heritability of complex traits could be due to genes that exert their influence at the level of intermediate phenotypes, such as arterial hypertension. However, they would not be powerful enough and contribute with sufficient effect to be detected at the level of the primary complex phenotype, such as acute myocardial infarction, in which they participate in its pathogenesis [12,17]. This possibility is consistent with the heritability being missing because differences between many common variants cannot attain statistical significance in Genome-Wide Association Studies (GWAS) studies [18,19,20,21]. Therefore, it was hypothesized that genes lacking sufficient strength to be detected at the main trait level (as mediated pleiotropy of intermediate phenotypes) account for a proportion of the missing heritability of this complex trait [12].

It is predicted that common small-effect genes affecting a complex trait might be located throughout much of the genome. For example, between 71% and 100% of 1 Mb windows could contribute to the heritability of schizophrenia [19], and thousands of expression Quantitative Trait *Loci* (eQTLs) may control blood gene expression [22]. Additionally, mathematical models predict that between 0.1 and 1% of single nucleotide variants (SNVs) have a causal effect on most diseases studied [23]. These observations align with the omnigenic model that purports to explain missing heritability [24,25]. Thus, attributing missing heritability to the genetic determinants that influence numerous intermediate phenotypes of a complex disease would require thousands of genes acting on its phenotypic variation and susceptibility [12]. It is difficult to identify some of these genetic markers that also would help predict the risk of complex trait diseases. Efforts have been made to develop genome-wide polygenic scores based on thousands of genetic variants [23]. However, the initial search for genetic determinants associated with intermediate phenotypes can be simplified in models of limited genetic variability, such as crosses of genetically homogeneous mouse strains [26,27].

In this work, we propose that genetic determinants linked to intermediate molecular phenotypes of CDA could help quantify susceptibility to this chemotherapy complication. We illustrate the rationale of our study in Figure 1 and Appendix A. Therefore, we assess the degree of CDA in a cohort of genetically heterogeneous mice generated by backcrossing. To identify intermediate molecular phenotypes of CDA, we quantify the myocardial levels of signaling pathways, microRNAs (miRNAs), and telomere length and evaluate their association with histopathological heart damage after chemotherapy. We then demonstrate that the genetic determinants associated with the levels of some of these intermediate molecular phenotypes in the myocardium contribute to the heritability of CDA. To do so, (i) first, we identify quantitative trait loci (QTLs) linked to the CDA intermediate molecular phenotypes (ipQTLs); (ii) second, we show that ipQTLs integrated into genetic models with a QTL directly linked to myocardial damage (cda-QTL) explain a more significant proportion of the phenotypic variability than does the cda-QTL alone. We conclude that since ipQTLs are not directly linked to CDA, they must contribute to its missing heritability. Thus, genetic determinants influencing the levels of intermediate molecular phenotypes in the myocardium, including the genes encoding the intermediate molecular phenotypes themselves, would contribute to the heritability of CDA. Subsequently, genes encoding intermediate molecular phenotypes of CDA may be markers of susceptibility to this side effect of chemotherapy. We evaluate this possibility in three cohorts of human patients.

## 2. Materials and Methods

### 2.1. Mouse Generation and Chemotherapy

We generated a genetically heterogeneous mouse cohort by backcrossing two inbred strains. We crossed a breast cancer-resistant mouse strain, C57BL/6 (hereafter C57), with a susceptible strain, FVB/J, to generate F1 mice. To generate the backcross mice (F1BX, after that), the non-transgenic F1 mice were crossed with *FVB/N-Tg(MMTVneu)202Mul/J* transgenic mice (hereafter FVB), carrying the *Avian erythroblastosis oncogene B2/Neuroblastoma-derived* (*ErbB2/cNeu)* protooncogene, expressed under the mouse mammary tumor virus (MMTV) promoter (MMTV-*Erbb2/Neu* transgene) and allowed to develop breast cancer as shown by Castellanos et al. [17,28].

Each mouse from the backcross cohort carried a unique combination of alleles from the two strains (FVB and C57) in variable proportions. In this combination, the genetic component from the FVB strain was predominant since it was used to generate the backcross with the F1 mice. FVB alleles can be homozygous or heterozygous, while the C57 component is reduced and heterozygous when present. Because each mouse is genetically unique, we can explore the phenotype variation (heart damage) among individuals throughout the cohort [29,30,31]. Thus, the mice were administered chemotherapy once they had developed breast cancer under isofluorane anesthesia and were monitored every two days. Mice were euthanized with CO_2_ when the tumors were bigger than 15 mm or showed signs of suffering, such as weight loss, dyspnea, and skin wounds. FVB transgenic mice were obtained from the Jackson Laboratories, and wild-type FVB/N and C57BL/6 mice were purchased from Charles River.

All mice were housed in ventilated filter cages in the Animal Research Facility of our Institution under specific-pathogen-free (SPF) conditions, environmental enrichment, and fed and watered ad libitum. We evaluated cardiotoxicity in 164 mice: 130 F1BX, 18 FVB, and 16 F1 (the latter having been generated after crossing FVB transgenic mice with C57). One group (*n* = 87, from them 70 were F1BX mice) was treated with doxorubicin every ten days with a dose of 5 mg/kg, and another group (*n* = 77, from them 60 were F1BX mice) received the combined therapy of doxorubicin (Pfizer) (5 mg/kg) plus docetaxel (Sanofi Aventis) (25 mg/kg), administered intraperitoneally every ten days. The age of the FVB and F1 mice was also heterogeneous because they were treated when the animals developed breast cancer, thus varying between 51 and 114 weeks of age. Without therapy, FVB control mice were used to evaluate subendocardial (*n* = 5) and subepicardial zones (*n* = 5). Litters from the same cage and age were randomly ascribed to each treatment group. The drug doses used mimicked the clinically relevant drug concentration [32].

In this study, the mice underwent four or five cycles of therapy, provided the chemotherapy was well-tolerated. Mice with less than four cycles were omitted. Once the treatment had finished, the mice’s evolution and tumor development were assessed every day for two months. Necropsies were then performed, and the heart and other tissues were collected and heart damage evaluated. All practices were previously approved by our Institution’s Institutional Animal Care and Bioethics Committee and conformed to the guidelines from Directive 2010/63/EU of the European Parliament on animal protection for scientific purposes. Thus, a protocol for treating mice was previously submitted and registered to our Institution Committee.

### 2.2. Heart-Tissue Processing and CDA Quantification

Hearts were fixed in 4% paraformaldehyde (Scharlau FO) for 24 h and then processed in an automatic system (Shandon Excelsior, Thermo Fisher Scientific, Waltham, MA, USA). The subsequent samples were sectioned, embedded in paraffin, and stained with hematoxylin-eosin with a standard protocol or the Masson-Goldner Trichrome kit (Bio-Optics, Daejeon, Republic of Korea) to evaluate the cardiac fibrosis and cardiomyocyte area. We robotically quantified heart fibrosis and the average area of myocardial fibers as pathophenotypes of CDA using the Ariol slide scanner to avoid intra- and inter-observer deviations. Histopathological damage was measured in the subendocardium and subepicardium from five randomly chosen regions of each sample. Regarding protocols for quantifying intermediate molecular phenotypes, see the Appendix A section.

### 2.3. Quantification of Signaling Proteins via Multiplex Bead Arrays

The quantification of signaling proteins within mouse heart protein lysates was achieved using multiplex assays. These were based on Millipore Luminex xMAP^®^ technology (Milliplex^®^, Burlington, MA, USA) and conducted in accordance with the manufacturer’s guidelines on a Bioplex 200 device (BIO-RAD, Hercules, CA, USA). Protein quantification adhered to specifications set forth by the manufacturer (Merck, Rahway, NJ, USA), with each sample containing 18.5 µg of total protein. The fluorescence readings from the heart samples fell within the linear dynamic range. Using a 7 Plex DNA Damage/Genotoxicity Magnetic Bead Kit (Milliplex Map Kit #48-621MAG, Millipore, Burlington, MA, USA), we quantified levels of ATR (total), CHK1(pS345), CHK2(pT68), H2AX(pS139), P53(pS15), MDM2 (total), and P21 (total). Buffer-compatible MAPmates assays were included to measure levels of activated caspase-3 and β-tubulin (#46-713MAG). A separate test employed a 9 Plex Multi-Pathway Magnetic Bead Kit (Milliplex Map Kit #48-680MAG) to quantify ERK1/2(pT185/pY187), P38(pT180/pY182), NFκB(pS536), JNK(pT183/pY185), AKT(pS473), P70S6K(pT412), CREB(pS133), STAT3(pS727), and STAT5(pY694/699). Lastly, the Milliplex MAP TGFβ Magnetic-Bead 3 Plex Kit (#TGFMAG64K-03, Millipore) was utilized to quantify TGFβ-1, TGFβ-2, and TGFβ-3 levels.

### 2.4. miRNA Quantification via QPCR

The miRNAs assess comprised miR-21a-5p, miR-29b-3p, miR-34a-5p, and the let-7 and miR families. Heart tissue miRNAs were analyzed using the 96:96 Fluidigm BioMark platform. A FAM-labeled, assay-specific TaqMan fluorescence probe Mix (Thermo-Fisher, Waltham, MA, USA) was employed, with each sample measured in triplicate. The QPCR protocol was as follows: 6.7 µL of the sample and 0.49 µL of the master mix were subjected to an incubation cycle at 50 °C for 2 min, followed by 70 °C for 30 min, and 25 °C for 10 min. This was succeeded by 50 °C for 2 min and 96.5 °C for 10 min, for 40 cycles in total. The final step consisted of 15 s at 96 °C and 1 min at 60 °C. Data analysis and cycle threshold (CT) value determination were facilitated by BioMark (Singapore) real-time PCR analysis software (Fluidigm Corp., San Francisco, CA, USA). Quantification followed the 2^−ΔΔCt^ method as proposed by Livak and Schmittgen in 2001 [33].

### 2.5. Mouse QTL Genetic Analyses and Generation of Genetic Models

Regarding mouse genotyping, see the Appendix A section. Identifying QTLs associated with fibrosis or cardiomyocyte area (cda-QTLs) and QTLs linked to intermediate molecular phenotypes in the myocardium (ipQTLs) were conducted with the same protocol via linkage analyses. Linkage analysis was carried out using interval mapping with the expectation-maximization (EM) algorithm and R/QTL software. The criteria for significant (lod score > 3) and suggestive (lod score between 1.5 and 2.99) linkages for single markers were chosen based on the findings of Lander and Kruglyak [34]. In the QTL results tables, the cXX.loc.XX markers do not refer to real SNVs but to genetic locations where the conditional genotype probabilities for the EM algorithm were calculated using the calc.genoprob function in R/qtl, with a step of 2.5 and an error.prob of 0.001. We estimated the number of mice needed for QTL analyses based on Sen and colleagues [35]. Various QTL models were developed in which the fitQTL function was used with Haley-Knott regression in R/qtl to fit and compare their LOD scores and the percentage of the explained variance [36]. QTLs were chosen for inclusion in the final model only if they demonstrated a significant additive or interaction effect (*p* < 0.05), determined via a “drop-one-QTL-at-a-time” analysis, which evaluates the impact of single QTLs or interactions.

### 2.6. Patients

The association of genetic variants with CDA was evaluated in three patient cohorts previously published. In the first two cohorts, comprising 71 anthracycline-treated pediatric cancer patients [37] (Paediatric Cohort) and 420 breast cancer patients (Breast Cancer Cohort) [38], cardiac function was assessed via echocardiography to evaluate the left ventricular fractional shortening (LVFS) or the left ventricular ejection fraction (LVEF), respectively. In the third cohort, cardiac magnetic resonance (CMR) was carried out in 24 cancer patients (CMR cohort) [39] at baseline and after every two cycles of a regular course of anthracycline therapy. All patients received anthracyclines in their treatment. Within the pediatric cohort, all patients underwent treatment utilizing either doxorubicin, daunorubicin, or epirubicin, conforming to their chemotherapy protocol [37]. In contrast, for the cohort of women diagnosed with breast cancer, epirubicin was the therapeutic agent of choice [38]. The cohort evaluated using CMR comprised breast cancer patients treated with epirubicin, whereas hematological patients received doxorubicin. Their clinical features have already been published [37,38,39]. Following the Declaration of Helsinki, the Bioethics Committee’s permission and the informed consent of the patients or their relatives in the case of pediatric patients were previously obtained for these studies.

### 2.7. Cardiac Magnetic Resonance: Acquisition and Analysis

Cardiac magnetic resonance (CMR) examinations were conducted with a Philips 1.5-Testa Achieva whole-body scanner (Philips Healthcare) equipped with a 16-element phased-array cardiac coil [39]. The imaging protocol always included a standard segmented cine steady-state free-precession (SSFP) sequence to provide high-quality anatomical references. The imaging parameters for the SSFP sequence were as follows: 280 × 280 mm field of view, 8 mm slice thickness with no gap, 3 ms repetition time, 1.50 ms echo time, 60° flip angle, 30 cardiac phases, 1.7 × 1.7 mm voxel size, and a single excitation. CMR images were analyzed using dedicated software (MR Extended Work Space 2.6, Philips Healthcare, Best, The Netherlands) by two observers experienced in CMR analysis and blinded concerning time-point allocation and patient identification.

### 2.8. Human Genetic Analyses

The existence of associations of CDA, measured using echocardiography or CMR, with SNVs, was evaluated in the three patient cohorts. We looked for alleles encoding proteins with levels in the myocardium that were associated with CDA in mice. SNVs were genotyped using the Infinium™ Global Screening Array-24 v2.0 BeadChip. Data were imputed using the Michigan Imputation Server with Minimac4 [40]. After retrieving the data, all markers with *R^2^* < 0.7 were removed from the analysis before proceeding further. Data were analyzed in R v3.6.0 (1). The SNPassoc package v2 was used to explore associations between mutations. Employing the “association” function, we performed case/control analyses of the possible genetic models (codominant, dominant, recessive, overdominant, and log-additive) to examine the associations between phenotypes and input mutations. SNVs from allelic forms of the genes encoding intermediate molecular phenotypes associated with myocardium damage in mice were used to generate prognosis risk scores in humans. A penalized restrictive analysis was carried out with the Least Absolute Shrinkage and Selection Operator (LASSO) machine-learning multivariate regression to reduce the risk of identification of false positives.

### 2.9. General Statistical Analyses

Pearson or Spearman correlation coefficients, depending on whether a Kolmogorov–Smirnov test indicated that the data were normally or non-normally distributed. The *t*-test, or the Mann–Whitney U test, was used to compare pairs of groups based on the distribution type. *p* < 0.05 was considered significant. Multiple regression was applied to determine which intermediate molecular phenotypes were associated with the definition of global heart fibrosis and the global cardiomyocyte area in mice. To construct the multivariate models, those variables (intermediate molecular phenotype levels in the myocardium associated with CDA (heart fibrosis or cardiomyocyte area) with *p* < 0.05) were chosen.

Principal component analyses (PCAs) were performed on the data with the function “prcomp” of “stats” R package. The results of the PCA analysis were plotted with “fviz_pca_biplot” function (factoextra” Rpackage), showing biplots of individuals and variables. The specific test conducted in each study is indicated in the legend of each figure or table. General statistical analyses were conducted using GraphPad Prism5, JMP 12, and R v3.6.0 [41]. The SNPassoc package [42] was used to analyze SNP associations.

## 3. Results

### 3.1. Cardiotoxicity Due to Anthracyclines Behaves as a Complex Trait in a Genetically Heterogeneous Mouse Cohort

CDA is a complex trait, and as such, identifying the genetic component in humans that influences CDA susceptibility is a challenging task [9]. However, crosses between syngeneic mouse strains simplify the identification of genetic determinants of complex traits [17,26,43]. Therefore, we generated a cohort of mice with a heterogeneous genetic background by backcrossing to identify genetic determinants linked to CDA susceptibility. We crossed MMTV-Erbb2/Neu transgenic mice with FVB background with F1 non-transgenic mice to generate the backcrossed cohort (hereafter, F1BX). These mice were treated with doxorubicin or combined therapy. Since doxorubicin and docetaxel are used in human cancer chemotherapy [44,45], the mice received treatment once they had developed breast cancer induced by the MMTV-Erbb2/Neu transgene [28] (Figure 2A).

Previous studies showed that, after anthracycline chemotherapy, a subclinical injury could be detected at the histopathological level in the myocardium even before functional damage had occurred [46,47]. Anthracyclines induce the death of cardiomyocytes that are replaced by fibrosis, leading to the atrophy of the left ventricle. Also, the atrophy of cardiomyocytes secondary to the toxicity of anthracyclines is described, which can be observed early using CMR [48]. However, in the long term, there is hypertrophy of the remaining cardiomyocytes due to ventricular remodeling secondary to diastolic overload when heart failure occurs [49,50,51]. Thus, variable grades of cardiomyocyte hypertrophy, myocytolysis, and fibrosis are characteristic features of ventricular remodeling associated with anthracycline exposure [52]. Thus, interstitial fibrosis and cardiomyocyte area modification are the phenotypes of pathological cardiac remodeling and chronic CDA [50,51]. We quantified both pathophenotypes in the myocardium after chemotherapy using an Ariol slide scanner to evaluate the degree of CDA, considering the heart’s global, subendocardial, and subepicardial zones. Heart damage was observed without chemotherapy (Figure 2A and Appendix A). Regarding age-related and intra-strain changes in CDA, F1 mice exhibited more frequent significant changes in CDA compared to FVB mice, indicating their higher sensitivity to anthracycline-induced cardiotoxicity (Appendix A).

Initially, we explored and compared heart fibrosis and the cardiomyocyte area between FVB and F1 mice. F1 mice treated with combined therapy had significantly higher levels of global (*p* = 0.0149) and subendocardial (*p* = 0.0043) fibrosis than did FVB mice (Figure 2B,C); we did not find more differences between both strains (Appendix A). We evaluated CDA in the F1BX genetically heterogeneous cohort of mice generated by backcrossing. As expected, the observed degree of cardiotoxicity spanned a wider range than seen in the parental strains and was distributed as a continuum throughout the F1BX mice (Figure 2B,C), as is characteristic of complex traits [53]. We then compared the CDA after doxorubicin treatment and combined therapy in the F1BX mice. Cardiotoxicity was higher after the combined therapy for global (*p* = 0.0066), subendocardial (*p* = 0.0083), and subepicardial fibrosis (*p* = 0.0546) than when doxorubicin was administered alone (Figure 2D–F); we did not observe differences in the cardiomyocyte area between both regimes of therapy (Appendix A). Globally, cardiotoxicity was more significant in the subendocardial than in the subepicardial area of the heart (Figure 2G–J). As expected, the combined therapy was more cardiotoxic than therapy with doxorubicin alone (Appendix A).

Chronic CDA susceptibility increases with age in humans [54], so we evaluated how heart damage varied with mouse age. Heart fibrosis increased with age after combined therapy (*p* = 0.0016) but not significantly after therapy solely with doxorubicin in global heart and subepicardial and subendocardial zones (Figure 2K–N and Appendix A). We divided the cohort into young and old groups of mice based on the sample’s median age of 71 weeks. Fibrosis increased with age in the old group after doxorubicin alone (*p* = 0.0052) and the combined treatment (*p* = 0.021) but not in young mice (Appendix A). We did not observe differences in cardiomyocyte area (Appendix A). Together, these results indicate that older mice were more sensitive to chronic CDA than younger mice, as previously observed in cancer patients [54].

The behavior and distribution of chronic CDA in F1BX mice as a complex trait are like those found in humans. Indeed, CDA was significantly greater after combined therapy than with doxorubicin alone [55,56] and with increasing age [54,57] and was distributed as a complex trait in the F1BX mice [53]. All these similarities justified using the F1BX backcross model to identify the genetic background component linked to chronic CDA.

### 3.2. Intermediate Phenotype Levels of Molecular Origin in the Myocardium Are Associated with Chemotherapy-Induced Cardiotoxicity

CDA is a complex trait. As such, its pathogenesis is influenced by different intermediate phenotypes at the systemic, tissue, cellular, and molecular levels [12,58]. We used this mouse backcross strategy as a simplified model to seek intermediate molecular phenotypes associated with CDA. We quantified the levels of a number of molecules in the myocardium following chemotherapy (as detailed in the methods section) and assessed their association with cardiac fibrosis and the area of the cardiomyocytes (Figure 3A). The molecules were selected based on their involvement in the pathogenesis of cardiomyopathy, as described in previous reports. Indeed, using multiplex bead arrays, we quantified levels of the myocardium proteins involved in antigenotoxicity pathways and cell-signaling pathways that favor or inhibit heart damage caused by anthracycline [59,60,61]. We also used qPCR to determine the miRNAs involved in cardiac diseases and cardiotoxicity [62] and in controlling myocardium telomere length [63].

The levels of intermediate molecular phenotypes involved in the pathogenesis of complex traits should be statistically significantly associated with the complex trait [14]. Indeed, some intermediate molecular phenotypes were associated with fibrosis variation and cardiomyocyte area in the F1BX mice (Appendix A). The molecular intermediate phenotypes that correlated with the CDA in the different conditions, collected in Appendix A (single or combined treatment, subepicardial or subendocardial area), were integrated into a principal component analysis and biplot representation. In this way, it was observed how these variables constituted by the molecular intermediate phenotypes helped to differentiate the groups of mice that had greater or lesser cardiotoxicity in terms of a greater or lesser degree of fibrosis (Figure 3B,D–G) and cardiomyocyte area (Figure 3H,I). We subsequently used multiple regression to evaluate which intermediate phenotypes were most important for defining CDA (Appendix A). For instance, after doxorubicin chemotherapy, young mice with low levels of P70S6K(pT412) and old mice with low levels of H2AX(pS139) in the myocardium had higher global fibrosis in the heart. After combined therapy, young mice with high levels of CREB1(pS133) presented high global fibrosis in the myocardium. AKT1(pS473), P38MAPK(pT180/pY182), β-tubulin and TP53(S15), and the miRNAs, miR210_3p, mR215_5p, Let7d_5d, and Let7d_5p, were associated with CDA under a variety of conditions. These selected molecular intermediate phenotypes also permit the classification of mice with high and low CDA susceptibility using Principal Component Analyses (PCA) (Appendix A). Furthermore, we conducted multivariate analyses (presented in Appendix A) to incorporate the degree of age participation in the context of CDA. The results indicate that age correlated positively with cardiac fibrosis and that the effect of age was particularly significant after the combined therapy (Appendix A).

In summary, these molecules associated with heart fibrosis and the cardiomyocyte area after chemotherapy were related to the chronic CDA susceptibility variation in the F1BX cohort of mice.

### 3.3. Identification of Genetic Determinants Linked to Intermediate Molecular Phenotypes of CDA

It has been indicated that genetic determinants linked to the intermediate phenotypic function of a complex trait could account for some of the phenotypic variations in the latter and contribute to its missing heritability [12,14]. Among the genetic determinants that determine the functional activity of an intermediate molecular phenotype, there are fundamentally those that regulate its levels. These determinants also include the gene that encodes the molecule with its regulatory sequences in cis and another series of genes in QTL regions located in trans that helps regulate molecular levels and activity [13,22], which we call intermediate phenotype QTLs (ipQTLs).

Following on, we asked whether the ipQTLs associated with intermediate molecular phenotypes of CDA contribute to the phenotypic variation of the latter. We set about integrating the ipQTLs with directly linked QTLs into genetic models with CDA (cdaQTLs) to determine whether they could account for more of the CDA phenotype variation than that explained solely via cda-QTLs. Accordingly, we looked for ipQTLs and cdaQTLs in the F1BX genetically heterogeneous mice that could subsequently be used in the genetic models (Figure 1 and Appendix A). Thus, firstly, we looked for the genetic regions (ipQTLs) linked with the myocardium levels of the intermediate molecular phenotypes, previously found to be associated with CDA in mice, via linkage analyses (Figure 4A and Appendix A). The global scenario of ipQTLs identified is shown as a heatmap (Figure 4B), and the specific information for each genetic locus is presented in Appendix A.

### 3.4. Identification of Genetic Determinants Directly Linked to CDA (cdaQTLs)

Following on, we searched for cdaQTLs associated with heart fibrosis and the cardiomyocyte area based on the type of chemotherapy (anthracycline alone or combined therapy) in different conditions: heart zone (whole heart, subendocardium, or subepicardium) and age (young or old mice) (Figure 5A). QTLs linked to CDA in these conditions were represented as heatmaps to visualize the global scenario of the genetic regions linked to heart damage after doxorubicin or combined therapy (Figure 5B,C and Appendix A).

Eighty cda-QTLs were identified, but some were in the same chromosome (Chr.) and were genetic regions simultaneously linked to several CDA conditions. In the end, we identified 27 cda-QTLs in full (Appendix A). For example, the same QTL was sometimes associated with the degree of fibrosis and the cardiomyocyte area under different conditions; this was the case of cda-QTL6 on Chr. 4 after doxorubicin treatment and cda-QTL11 on Chr. 10 after combined therapy (Figure 5D and Appendix A). The same QTL and pathophenotype were occasionally associated in both chemotherapy regimens, e.g., the cda-QTL13 on Chr. 11 and heart fibrosis (Figure 5E and Appendix A). Notably, the cda-QTL6 on Chr. 4 was explicitly associated with CDA in old mice, whereas the cda-QTL13 on Chr. 11 was most frequently related to CDA susceptibility in young mice (Figure 5D,E and Appendix A). Identifying multiple QTLs associated with CDA confirmed the polygenic component of susceptibility to this complication, even in a simplified model like that of the F1BX mouse cohort [10].

### 3.5. ipQTLs Integrated into Genetic Models with cdaQTLs Account for More Phenotypic Variation of CDA Than Explained by cda-QTLs Alone

Our next step was integrating the ipQTLs with cda-QTLs into the genetic models [36] to evaluate whether these could account for more of the CDA phenotype variation than that explained solely by the cda-QTL (Figure 6A). In doing so, we wanted to demonstrate that these ipQTLs contribute to the missing heritability of the CDA. The cda-QTL would enable the ipQTLs contributing to the missing heritability of CDA to be revealed via the genetic models (Figure 1 and Appendix A). As examples of cdaQTLs linked with CDA in different conditions, we selected cdaQTL6 and cdaQTL13 (Figure 5D,E) to evaluate whether ipQTLs integrated with these cdaQTLs could account for more of the CDA phenotype variation than that explained solely by cdaQTL6 and cdaQTL13 (Figure 6A). First, we estimated the phenotypic variance of global fibrosis attributable to them in the F1BX cohort. CdaQTL6 explained 22.17% of the CDA variance in global fibrosis in old mice after doxorubicin chemotherapy, and cdaQTL13 accounted for 28.82% and 26.64% of the CDA variance in global fibrosis in younger mice treated with doxorubicin or combined therapy, respectively (Figure 6B).

We then assessed whether ipQTLs linked to myocardium molecules increased the amount of global heart fibrosis explained by cdaQTL6 in old mice and cdaQTL13 in young mice. The intermediate molecular phenotypes that were correlated with global fibrosis under these conditions and their ipQTLs are shown in Figure 6C,D and Table 1A. We examined all the viable genetic models with cdaQTL6 or cdaQTL13 and the ipQTLs linked to the intermediate molecular phenotypes associated with global fibrosis [36] (Appendix A).

Concerning global heart fibrosis in young mice, we observed that the phenotypic variation due to cda-QTL13 increased after including some of the ipQTLs associated with P70S6K in genetic models (Figure 6C and Table 1B). The ipQTLs linked to P70S6K levels in young mice were located on Chr. 3 (dox-ipQTL51), Chr. 4 (dox-ipQTL52), Chr. 9 (dox-ipQTL53), and Chr. 17 (dox-ipQTL54) (Table 1A). The variance in global fibrosis explained by cda-QTL13 was 28.82%. This increased to 47.55% when considering the dox-ipQTL51 (model 11), to 57.31% when including the dox-ipQTL51 and dox-ipQTL54 (model 17), and to 65.9% for dox-ipQTL51 and dox-ipQTL53 (model 18) (Figure 6C and Table 1B).

As indicated, the criteria for choosing intermediate molecular phenotypes of CDA were based on the evidence from previous studies. In the case of P70S6K, protective and anti-protective effects after treatment with doxorubicin have both been described [64,65,66]. We confirmed its role in CDA with functional in vitro studies. Human-induced pluripotent stem cell-derived cardiomyocytes (hi-PSC-CMs) are used to ensure the involvement of genes in cardiotoxicity at a functional level [67]. Hence, as an example, we demonstrated that downregulating *RPS6KB1* levels with siRNA in hiPSC-CMs increases their sensitivity to doxorubicin, confirming the role of P70S6K as an intermediate molecular phenotype of CDA (Appendix A).

Similarly, concerning global heart fibrosis in old mice, cdaQTL6 explained 22.17% of the phenotypic variation. This value was higher when ipQTLs associated with myocardium levels of ɤH2AX and pJNK were included in the models (Figure 6D and Table 1). The amount of phenotypic variation in CDA explained by genetic determinants directly linked to CDA (cdaQTLs) was boosted using ipQTLs linked to intermediate molecular phenotypes associated with CDA, implying that the genetic determinants controlling the intramyocardial levels of these intermediate molecular phenotypes contribute to the phenotypic variance of CDA. However, as they are QTLs directly linked to the CDA, we can deduce that they are the source of some of the missing CDA heritability. This large amount of phenotypic variation elucidated can be explained by the simplicity of the backcross model, with a more limited genetic diversity than human populations [68].

### 3.6. Genes Encoding Intermediate Molecular Phenotypes Associated with Myocardium Damage in Mice Can Be Genetic Determinants of CDA in Patients

The previous analyses showed that ipQTLs linked to the levels of intermediate molecular phenotypes of CDA increase the proportion of phenotypic variation explained by cda-QTLs. However, the levels of these intermediate phenotypes associated with CDA also depend on the regulatory regions of the genes encoding the intermediate phenotypes themselves. Given this, we can presume that all of them, the regulatory regions in cis with the gene encoding the intermediate phenotype, and the ipQTLs, mostly in trans, may contribute via the levels of the intermediate molecular phenotype to CDA variation. The ipQTL driver genes are unknown; however, it is reasonable to propose that the genes that encode the intermediate molecular phenotypes of CDA and control their levels can be genetic determinants of CDA susceptibility (Figure 1 and Appendix A).

We evaluated the extent to which allelic forms of the genes encoding intermediate molecular phenotypes associated with myocardium damage in mice could be genetic determinants of CDA in three cohorts of cancer patients treated with anthracyclines. CDA was evaluated via echocardiography in a cohort of women with breast cancer and another with pediatric cancer; in the third cohort, the CDA was evaluated via cardiac magnetic resonance (CMR), considering an LVEF reduction of 5% or more during the first six months or throughout the complete follow-up. We only evaluated SNVs from those genes that encoded molecules associated with CDA in mice, testing the most probably genetic model (Figure 3 and Appendix A). Several single nucleotide variants (SNVs) were associated with susceptibility to CDA in patients, some of which were noted in more than one cohort (Appendix A).

CDA susceptibility is partly different in adults and in pediatric patients [37,38,69]. Subsequently, we used the two largest cohorts, formed by breast cancer patients and pediatric patients, to generate two polygenic risk scores. Thus, each cohort was divided into a training set (80%) and a testing set (20%). In the training set, after bootstrapping 100 times, a series of SNVs associated with susceptibility to CDA were identified. Subsequently, two different risk models were obtained using a restrictive LASSO regression analysis (Figure 7). Interestingly, the SNVs that were part of the risk scores belonged to the same five genes in both models. Indeed, the CDA model in adults consisted of SNVs of *AKT1*, *STAT3*, *TP53*, *MAPK8*, *MAPK11*, and *RelA/P65*; the pediatric model consisted of SNVs of the same first five genes.

Our results highlight the value of genetic mouse models as tools for identifying the intermediate phenotypes that contribute to the variation of CDA and the use of the genes encoding them as potential susceptibility markers.

## 4. Discussion

Chronic CDA is a common side effect that can be very severe and affect the continuity of chemotherapy treatment [1]. CDA susceptibility varies considerably among patients [6], but significant efforts have been made, through the use of genetic markers, to identify the patients who are susceptible to developing chronic CDA [10]. A predisposition to CDA has a strong genetic component, and, as a complex disease or trait, it is, by definition, polygenically inherited [9]. The genetic elements of complex characters are difficult to identify; there are substantial discrepancies between the proportion of phenotypic variance expected to arise from genetic causes (expected heritability) and the heritability explained by identified DNA sequencing variants. The difference between them is known as missing heritability [11]. CDA, being a complex trait, has an unknown degree of missing heritability, making it challenging to identify most of its genetic components.

The use of intermediate phenotypes to identify a part of the genetic component of complex traits has been proposed in psychiatric disorders [14] and extended to other fields [12]. It has been suggested that identifying the genetic determinants associated with intermediate phenotypes essential for complex-trait pathogenesis could help identify a part of the missing heritability and yield genetic markers for predicting complex disease susceptibility [12]. Part of the genetic component of the missing heritability could be explained by the genetic determinants, such as QTLs, that are linked to intermediate phenotypes involved in the pathogenesis of complex traits. Indeed, a highly influential QTL can be simultaneously detected at the intermediate phenotype and primary complex trait levels, reflecting a process known as mediated pleiotropy [70]. It has been proposed that QTLs that cannot be detected as mediated pleiotropy could be part of the missing heritability because they are too weak to be revealed via genetic analysis at the complex-trait level [12,17].

The network of intermediate phenotypes at the systemic, tissue, cellular, and molecular levels that determines the pathogenesis of a complex trait is regulated by a multitude of genes acting at all those levels. This scenario coincides with the omnigenic model, involving a series of gene networks with core genes and many peripheral genes that has recently been proposed to explain missing heritability [24,25]. It is inferred from this model that the complete heritability of a complex trait is controlled by most of the genome [19,23,25]. The source of most heritability is genes with so little effect that many are difficult, or even impossible, to identify, no matter how many individuals are studied.

From a practical point of view, the challenge is to find a way to determine the genetic markers of susceptibility to a complex trait. The focus on the genetic determinants associated with the variation of intermediate phenotypes (which, in turn, are associated with a complex trait) makes it possible to identify essential genes that can be susceptibility markers of diseases of complex genesis. In this sense, the approach proposed in this study could be adopted as a general strategy for better identifying genetic markers of high-prevalence complex diseases, e.g., type II diabetes, autoimmune diseases, thrombosis, cardiovascular diseases, sporadic cancers, and CDA [71].

It is difficult to determine the polygenic component of complex human population traits because of their genetic complexity and sophisticated environmental interactions [26]. Identifying genes with a weak effect in human studies using techniques such as GWAS is complicated because enormous sample sizes are required to demonstrate statistical significance. In addition, the massive amount of multiple testing supposed by the analysis of millions of SNVs dramatically reduces statistical power, especially when trying to locate common genetic variants of weak effect. However, quantifying the phenotypic variation of complex traits, under controlled environmental conditions, in a simplified genetic model consisting of crosses between genetically homogeneous strains of mice can guide the choice of candidate genes and pathways to be tested in human populations. Identifying candidate genes in this simplified genetic model reduces the number of genetic variants that need to be considered [27]. We think this strategy makes it possible to identify genetic markers of complex traits without carrying out studies in thousands of patients because evaluating intermediate molecular phenotypes in the simplified genetic model in mice enables the selection of candidate genes.

This work has identified some genetic components linked to intermediate molecular phenotypes associated with CDA in a mouse backcross model, which helped identify some genetic elements related to the CDA susceptibility itself. Indeed, variable susceptibility to doxorubicin-induced cardiotoxicity observed in humans can be modeled in a panel of the collaborative cross strains [72]. Since anthracyclines exert their toxicity by damaging DNA, CDA intermediate phenotypes would differ regarding the molecular pathways involved in DNA damage response or in the signaling pathways, such as AKT [59] and P38MAPK [60], that promote or protect from heart damage by anthracyclines [61]. We also hypothesize that other CDA intermediate phenotypes can be molecules involved in heart diseases and cardiotoxicities, such as cell signaling pathways [61], miRNAs [62], and telomere length [63]. Multiple regression models allowed us to pinpoint which of these intermediate molecular phenotypes, determined in mouse myocardium, can best explain the phenotypic variation in the CDA. We then evaluated the extent to which the ipQTLs associated with these intermediate molecular phenotypes account for chronic CDA.

Although ipQTLs were not directly linked to CDA, we used genetic models to demonstrate that these ipQTLs account for a part of the heritability of the CDA. These ipQTLs help control in trans the levels of intermediate protein phenotypes in the myocardium. Protein levels depend on cis factors, the most important being the DNA regulatory sequences in the gene encoding the molecule and genetic elements in “trans.” We have used ipQTLs in –“trans” to demonstrate that genetic determinants linked to the levels of the molecule contribute to the phenotypic variation of CDA. Although we do not know which genes drive the effects of ipQTLs, we know the gene encoding the intermediate protein phenotype in *cis*. So, if genetic variants of these genes determine the protein intermediate phenotype levels, they will also contribute to the heritability of the complex trait. For this reason, we looked for variants of these genes to evaluate in the human population.

An enormous number of intermediate phenotypes affect the pathogenesis and variation in a complex trait like CDA, so, unsurprisingly, many of the contributing genetic determinants cannot be detected among those of the central phenotype, for which reason they are responsible for much of the missing heritability [12]. Furthermore, the myriad interactions between intermediate phenotypes and the abundance of QTLs associated with them make it unlikely that the sources of missing heritability of a complex phenotype, including CDA, could ever be accounted for completely. Indeed, these intermediate phenotype interactions at different levels may involve most of the genome [25].

In summary, a genetically diverse group of mice can be leveraged to uncover the genetic influences on protein levels in the myocardium associated with histopathological damage following chemotherapy. This process will shed light on elusive genetic factors related to CDA in both mice and humans. By pinpointing the genetic and molecular factors contributing to elevated CDA risk, we can enhance our capabilities in predicting and mitigating CDA. Our findings suggest that the proposed approach may broadly facilitate the identification of susceptibility markers for complex diseases. Furthermore, the genetic markers discovered could assist in identifying patients at heightened risk of developing CDA. This would enable more personalized patient care and optimized chemotherapy regimens, potentially reducing the risk of severe adverse drug reactions.

In terms of study limitations, further epidemiological investigations are required to ascertain the extent to which genetic determinants linked to intermediate phenotypes of a complex trait can aid in identifying useful genetic markers for risk prediction. Although these genetic determinants might not be potent enough to be identified as associated with the primary complex phenotype, they should exert a sufficient impact to contribute to risk explanation within a polygenic model context. Such genetic influences could help elucidate the likelihood of disease manifestation.

## Figures and Tables

**Figure 1 cells-12-01956-f001:**
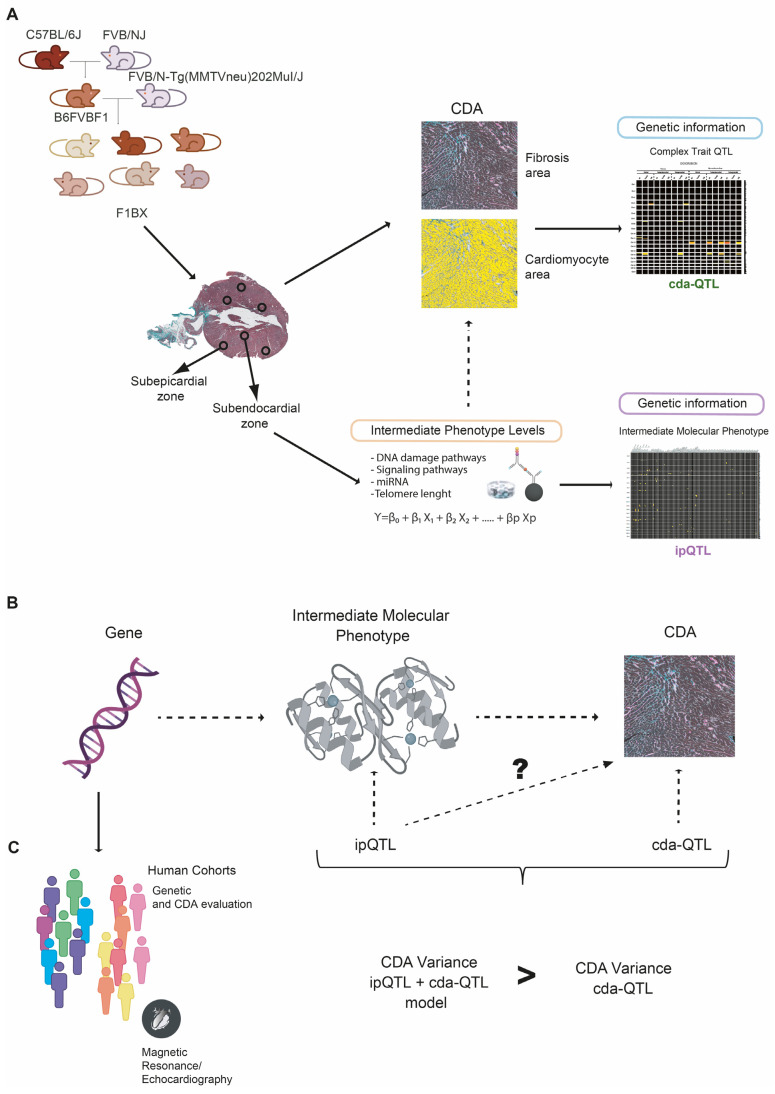
Overview of the general approach. (**A**) This study investigates variations in susceptibility to Anthracycline-Induced Cardiotoxicity (CDA) within a backcrossed mouse cohort. CDA severity is quantified histopathologically using an Ariol slide scanner to assess both the subepicardial and subendocardial regions of the heart. Further, we quantify the myocardial levels of various intermediate phenotypes associated with cardiotoxicity. Following this, we identify genetic regions or Quantitative Trait Loci (QTL) correlated with CDA (cdaQTL), and other QTLs linked to intramyocardial levels of intermediate molecular phenotypes (ipQTL). (**B**) Subsequently, we examine whether certain ipQTLs significantly contribute to the phenotypic variation (susceptibility) of CDA, even if they lack a direct link. We employ genetic models to evaluate whether the phenotypic variation explained by the combined ipQTL and cdaQTL surpasses that explained solely by the cdaQTL. If true, the genetic determinant associated with myocardial levels of this intermediate molecular phenotype contributes to CDA susceptibility. (**C**) The genes responsible for ipQTLs remain unidentified. Nevertheless, any genetic determinant influencing levels of cardiotoxicity-associated molecules could potentially contribute to CDA susceptibility. Therefore, allelic variations of genes encoding these intermediate molecular phenotypes emerge as candidates for evaluation within the human population.

**Figure 2 cells-12-01956-f002:**
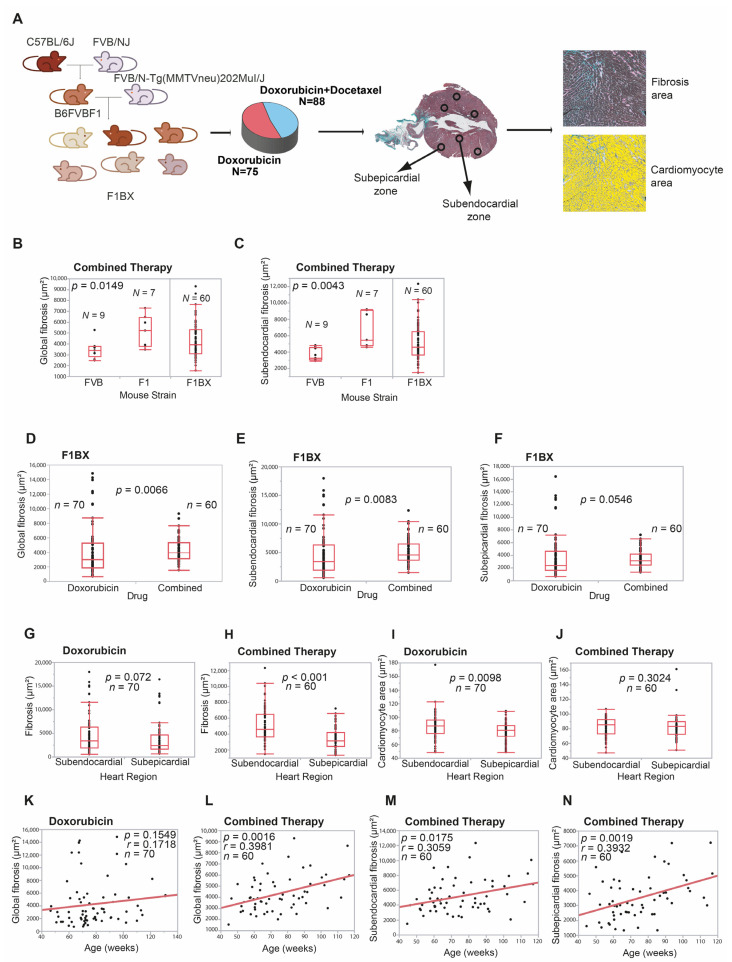
Genetic background, therapy regime, and age influence anthracycline-induced cardiotoxicity. (**A**) A backcrossed mouse cohort is used to evaluate Cardiotoxicity Due to Anthracyclines (CDA). Following chemotherapy treatment, cardiotoxicity is quantified histopathologically. (**B**,**C**) We compare cardiotoxicity between homogeneous genetic background mouse strains, FVB and F1. Following combined chemotherapy, FVB mice display reduced global heart fibrosis (**B**) and lessened subendocardial fibrosis (**C**) compared to F1 mice. No observable differences exist between the parental strains post-doxorubicin chemotherapy alone (Appendix A). Please note the continuous distribution of heart fibrosis in F1BX mice (**B**,**C**). (**D**–**F**) For the genetically diverse F1BX mouse cohort, combined treatment induces greater cardiotoxicity than doxorubicin alone, as evidenced by the degree of global (**D**), subendocardial (**E**), and subepicardial (**F**) fibrosis. (**G**–**J**) In F1BX1 mice, CDA is more prevalent in the subendocardial zone than in the subepicardial zone, with respect to fibrosis area (**G**,**H**) and cardiomyocyte size (**I**,**J**), as analyzed via the Mann–Whitney U test. (**K**,**L**) The relationship between age, type of chemotherapy, and CDA in the entire F1BX mouse cohort is examined. No correlation between global fibrosis and age is found with doxorubicin treatment alone (**K**). However, a positive correlation is observed after combined treatment (**L**), as estimated using the Spearman correlation coefficient. (**M**,**N**) With age, global fibrosis intensifies in older mice treated with either doxorubicin (**L**) or combined therapy (**M**), as determined using the Spearman correlation coefficient. Only statistically significant results are shown.

**Figure 3 cells-12-01956-f003:**
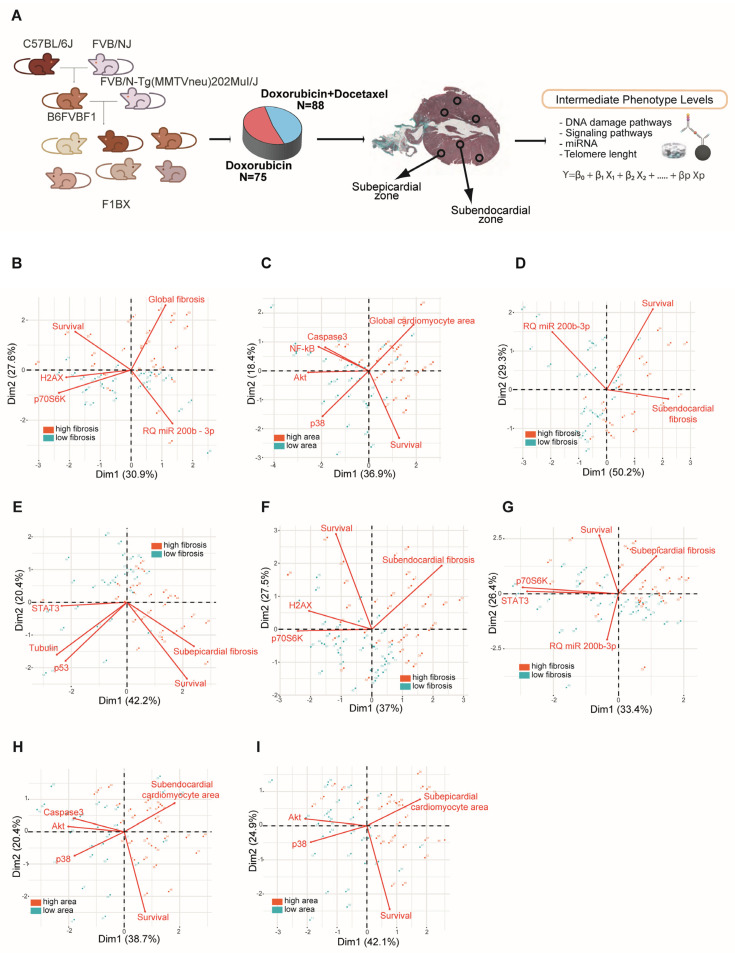
Association between intermediate molecular phenotypes and Anthracycline-Induced Cardiotoxicity (CDA) in F1BX mice. (**A**) Quantification of various intermediate molecular phenotypes in the myocardium of F1BX mice post-chemotherapy treatment; (**B**–**I**) Classification of mice into high and low CDA susceptibility groups via principal component analysis based on the myocardial levels of these phenotypes under different conditions: fibrosis following doxorubicin treatment (**B**), changes in cardiomyocyte size after combined therapy (**C**), subendocardial fibrosis post combined therapy (**D**), subepicardial fibrosis post combined therapy (**E**), subendocardial fibrosis following doxorubicin treatment (**F**), subepicardial fibrosis following doxorubicin treatment (**G**), changes in subendocardial cardiomyocyte size after combined therapy (**H**), and changes in subepicardial cardiomyocyte size after combined therapy (**I**). High (depicted in brown) and low (depicted in blue) levels of fibrosis and cardiomyocyte size, defined by the median, are used to differentiate the mice. This figure complements the data presented in Appendix A.

**Figure 4 cells-12-01956-f004:**
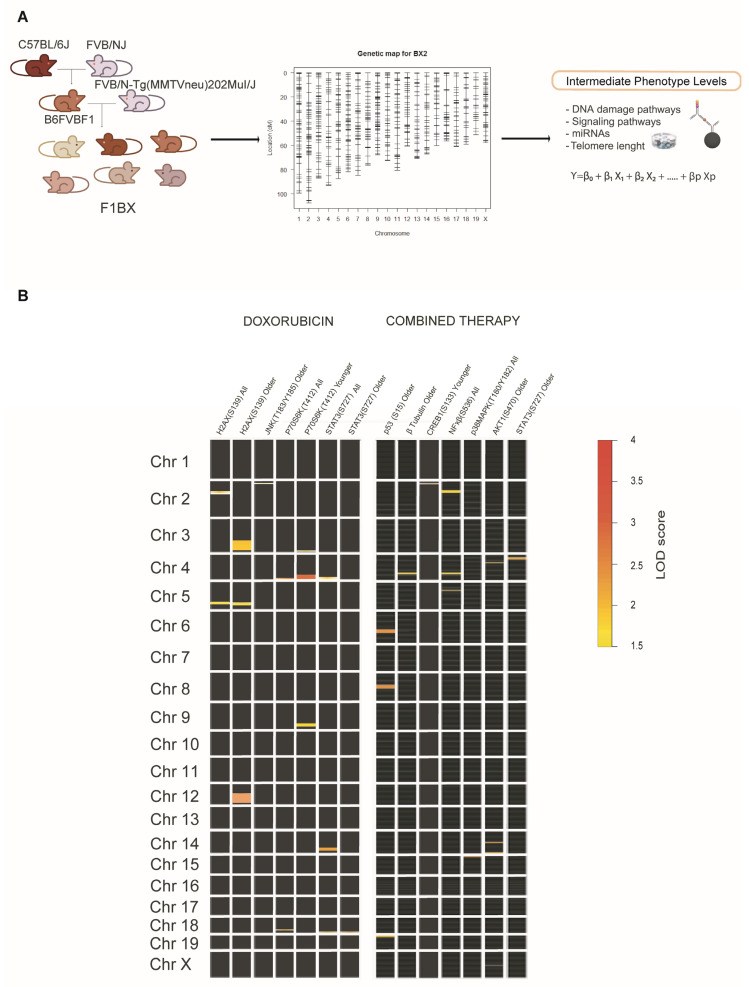
Identification of quantitative trait loci associated with Intermediate Molecular Phenotypes (ipQTLs) of CDA susceptibility. (**A**) F1BX1 mice, generated by backcrossing, were genotyped using 1499 SNVs on an Illumina platform, followed by linkage analysis to detect ipQTLs. (**B**) The heatmap illustrates the global distribution of ipQTLs post-doxorubicin or combined therapy. Each square denotes a chromosome, numbered on the left. The intensity of each square indicates the degree of association (LOD score) between the genetic markers and the phenotype, as per the provided scale. The position of the marks within each square corresponds to their relative location within each chromosome (centromere and telomere positions above and below, respectively). Only linkages with a LOD score > 1.5 (suggestive) are represented. R/qtl software was utilized to identify ipQTLs. The precise location of each ipQTL on each chromosome and the associated genetic markers are detailed in Appendix A (doxorubicin therapy) and Appendix A (combined therapy).

**Figure 5 cells-12-01956-f005:**
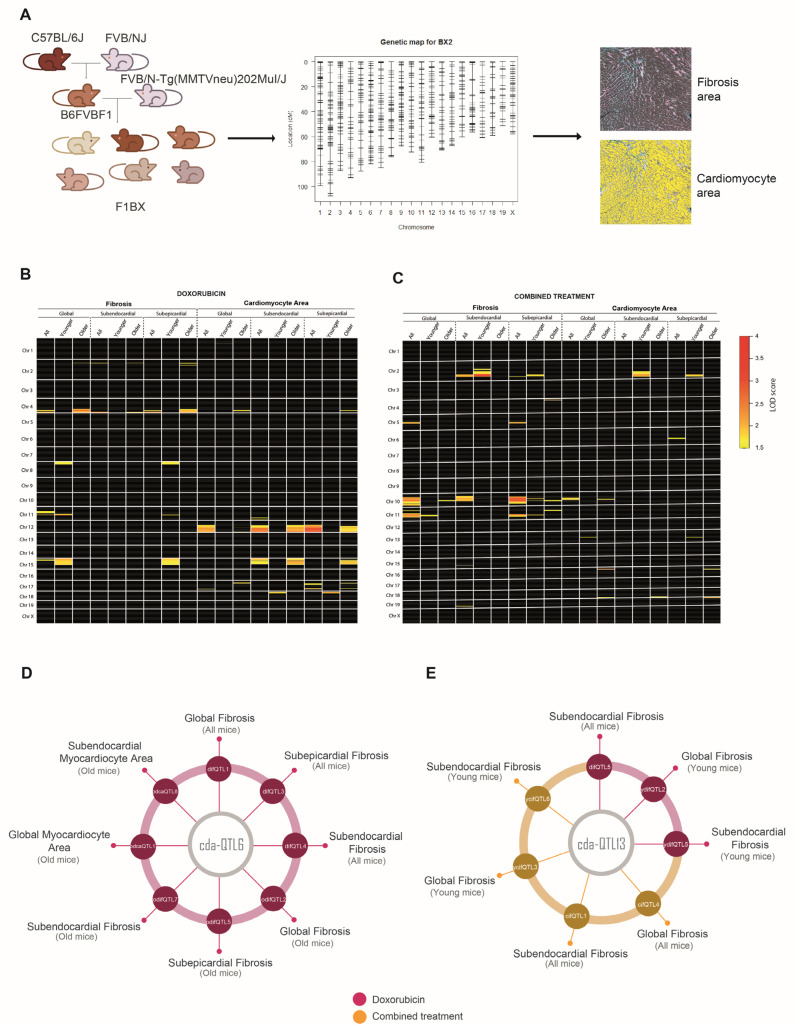
Identification of Quantitative Trait Loci associated with Anthracycline-Induced Cardiotoxicity (cda-QTLs). (**A**) Genotyping of the F1BX mouse cohort facilitated the localization of cdaQTLs; QTLs intrinsically connected to CDA susceptibility as determined via histopathological analysis. (**B**,**C**) The heatmaps provide a comprehensive depiction of the distribution of cda-QTLs associated with cardiac fibrosis and cardiomyocyte size under diverse conditions (age and therapy type): post-doxorubicin chemotherapy (**B**) or post-combined therapy (**C**). For both panels, (**B**,**C**), only cdaQTLs possessing a LOD score > 1.5 were charted on each chromosome. Analyses were executed using R/qtl software. Each square symbolizes a chromosome, numbered to the left. The intensity of the markings within each square represents the level of association (LOD score) between the genetic markers and the phenotype. The mark’s position corresponds to its relative location within the chromosome, with centromere and telomere positions represented above and below, respectively. The precise locations of each cda-QTL and their corresponding genetic markers can be found in Appendix A. (**D**) When identifying cdaQTLs in older mice under different conditions, we noted the repeated appearance of the same region, leading to the identification of cda-QTL6. Indeed, cda-QTL6 colocalized with multiple cda-QTLs: difQTL1 (doxorubicin-induced fibrosis QTL1), difQTL3, difQTL4, odifQTL 4 (older mice difQTL4), odifQTL5, odifQTL7, odcaQTL1 (older mice doxorubicin-induced cardiomyocyte area QTL1), and odcaQTL8. (**E**) A similar observation was made for cda-QTL13, which was associated with fibrosis in younger mice. Indeed, cda-QTL13 colocalized with the following cdaQTLs: difQTL5, ydifQTL2 (young mice difQTL2), ydifQTL5, cifQTL4 (combined therapy-induced fibrosis QTL4), cifQTL11, ycifQTL3 (young mice cifQTL3), and ycifQTL6. The numerical data (LOD scores and peak locations of markers) relating to the cdaQTLs detailed in panels (**C**,**D**) are provided in Appendix A.

**Figure 6 cells-12-01956-f006:**
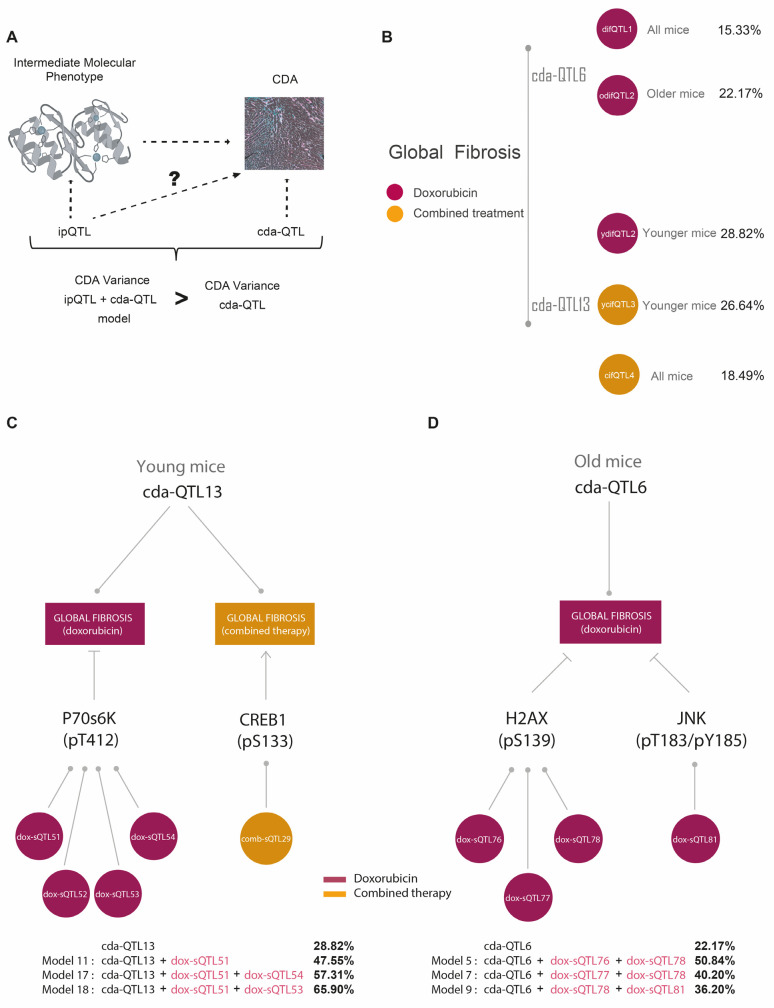
Percentage of global fibrosis explained by cda-QTL6 or cda-QTL13 and genetic models. (**A**) Genetic models between ipQTLs and cdaQTL are created to assess whether an ipQTL contributes to the phenotypic variation of CDA. If the model’s variability significantly exceeds that of the cdaQTL, the ipQTL would contribute to the phenotypic variation of CDA and susceptibility. (**B**) The diagram above shows the percentage of global fibrosis explained by cda-QTL6 post-doxorubicin treatment (red). The circles on the right denote the name of the cda-QTL6 under these conditions, difQTL1 (doxorubicin-induces fibrosis QTL1), which appears linked to global fibrosis in all mice, and odifQTL2 (old mice difQTL2), which seems to be associated with global fibrosis in all mice. Below is the percentage of global fibrosis explained by cda-QTL13 following treatment with doxorubicin (red) or combined therapy (yellow). The circles on the right show the other names of the cda-QTL13 for those conditions: ydifQTL2 (young mice doxorubicin-induced fibrosis QTL2), ycifQTL3 (young mice combined therapy-induced fibrosis QTL3), and cifQTL4 (combined therapy-induced fibrosis QTL3), the latter linked to global fibrosis in all mice. (**C**,**D**) We selected cda-QTL6 and cda-QTL13 to assess whether ipQTLs combined with these cda-QTLs could account for more of the CDA trait variation than that justified exclusively via cda-QTL6 and cda-QTL13. Thus, we selected cda-QTL6 and cda-QTL13 as examples to generate the genetic models—schemes illustrating the components used to develop the genetic models (Table 1). (**C**) In this case, the objective was to evaluate whether genetic models can explain more global cardiac fibrosis variance after doxorubicin treatment in young mice than cda-QTL13. To this end, the two intermediate molecular phenotypes associated with global fibrosis in young mice after treatment with doxorubicin were P70S6K(pT412) and CREB1(pS133) (Table 1A and Appendix A) and the ipQTLs linked with them are shown in Table 1A and Appendix A. The genetic models were developed with cda-QTL13 and four combinations of the ipQTLs linked to P70S6K(pT412) (dox-ipQTL51, dox-ipQTL52, dox-ipQTL53, and dox-ipQTL54), and one combination, comb-ipQTL29, linked to CREB1(pS133). The phenotypic variability explained by cda-QTL13 alone and the significant increase in those genetic models in which it improved is indicated below the diagram (Table 1B). (**D**) Scheme to illustrate the same results as panel (**C**) but for the case of cda-QTL6 and global fibrosis in old mice. Panels (**C**,**D**) correspond to Table 1.

**Figure 7 cells-12-01956-f007:**
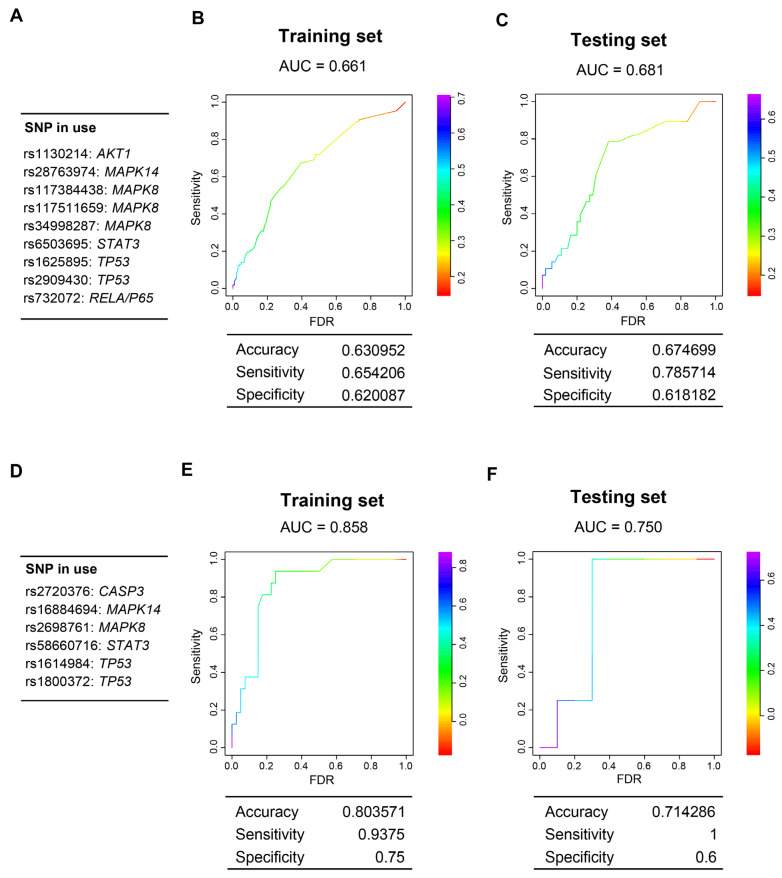
Genetic models of CDA risk generated by bootstrapping 100 times and LASSO multivariate regression. Each human cohort was randomly divided into 80% training and 20% testing sets. During model construction, logistic regression was employed in a bootstrapping strategy with a fixed sampling rate (80% evaluation and 20% validation) and numerous iterations (100). After bootstrapping, the SNVs, significantly associated with CDA in evaluation and validation data at least five times across 100 iterations, were selected to construct the Least Absolute Shrinkage and Selection Operator (LASSO) regression model on the 80% training set. After training, the best cutoff on the Receiver Operating Characteristic (ROC) curve was optimized based on the maximum Youden’s index (sensitivity + specificity − 1), and the LASSO models, along with the corresponding optimal cutoff, were applied to the 20% testing set for independent evaluation of the model performance. (**A**–**C**) CDA risk model for the cohort of breast cancer patients. The genetic model (**A**) and the ROC curves of the training set (**B**) and the testing set (**C**) are shown. (**D**–**F**) Genetic risk model for the pediatric patient cohort. The genetic model (**D**) and the ROC curves for the training set (**E**) and the testing set (**F**) are shown.

**Table 1 cells-12-01956-t001:** Genetic models. (**A**) The upper table presents the components utilized in the development of the genetic models, which include the intermediate molecular phenotypes associated with global fibrosis in both young and old mice (Appendix A) (**a**) and the ipQTLs linked to these intermediate phenotypes (Appendix A) (**b**). For additional clarification, refer to Figure 5C,D. N.I., not identified. (**B**) Enhancement of the CDA variation explained by cdaQTL6 or cda-QTL13 with ipQTLs integrated into genetic models. Genetic models that incorporate ipQTLs linked to intermediate molecular phenotypes increase the proportion of phenotypic variation of global fibrosis explained by cdaQTL13 in younger mice and cda-QTL6 in older mice treated with doxorubicin. Only models that improved the fibrosis phenotypic variability explained are included (Appendix A). Genetic models were generated using the Fitqtl function (r/qtl package). n.a., not applicable. This table is related to Figure 6C,D.

(A) QTLs and CDA Conditions Used in the Genetic Models					
cda-QTL	Therapy Type	Mouse Age Group	Pathophenotype of CDA	(a) Molecular Intermediate Phenotypes Associated with Global Fibrosis	(b) ipQTLs
**cda-QTL6**	Doxorubicin	Old Mice	Global Fibrosis	γH2AX(S139)	dox-ipQTL76
					dox-ipQTL77
					dox-ipQTL78
				JNK(T183/Y185)	dox-ipQTL81
				miR200b-3p	N.I.
**cda-QTL13**	Doxorubicin	Young Mice	Global Fibrosis	p70S6K(T412)	dox-ipQTL51
					dox-ipQTL52
					dox-ipQTL53
					dox-ipQTL54
	Combined Therapy	Young Mice	Global Fibrosis	CREB(S133)	comb-ipQTL29
**(B) Improvement of the Global Fibrosis Variation Explained by cda-QTL6 and cda-QTL13 with ipQTLs in Genetics Models**					
**Basal Effect/Model Effect**		**Model Components**	**LOD Score**	**Fibrosis Variation (%)**	***p*-Value**
**cda-QTL13**	**Basal effect**	n.a.	2.38	28.82	n.a.
**Models with cda-QTL13 in young mice**	**Model 11**	cda-QTL13; dox-ipQTL51	3.78	47.55	0.0006
	**Model 17**	cda-QTL13; dox-ipQTL51; dox-ipQTL54	4.99	57.31	0.002
	**Model 18**	cda-QTL13; dox-ipQTL51; dox-ipQTL53	6.3	65.90	0.0001
**cda-QTL13**	**Basal effect**	n.a.	2.29	22.17	n.a.
**Models with cda-QTL6 in old mice**	**Model 5**	cda-QTL6; dox-ipQTL76; dox-ipQTL78	6.63	50.84	0.00007
	**Model 7**	cda-QTL6; dox-ipQTL77; dox-ipQTL78	4.8	40.20	0.0024
	**Model 9**	cda-QTL6; dox-ipQTL78; dox-ipQTL81	4.2	36.20	0.0075

## Data Availability

This published article and its Appendix A files include most of the data generated and analyzed in this study. Related metadata underlying the findings are available as additional datasets in the public repository DIGITAL.CSIC http://hdl.handle.net/10261/239215 (accessed on 17 December 2022). The other human genetic and clinical data are available upon reasonable request from those of us who are the corresponding authors of previously published manuscripts [37,38,39].

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
