# Peer review of "Intermediate Molecular Phenotypes to Identify Genetic Markers of Anthracycline-Induced Cardiotoxicity Risk"

_cells, 2023, doi:10.3390/cells12151956_

Round 1

Reviewer 1 Report

cells-2378592

The authors propose that innovative research into the levels of intermediate molecular phenotypes (IMPs) in the myocardium associated with histopathological damage may explain sensitivity to CDA anthracyclines, and genes encoding IMPs to identify patients susceptible to this complication. To this purpose, a cohort of mice (n = 165) treated with doxorubicin and docetaxel were studied, quantified: cardiac fibrosis using an Ariol slide scanner and intramyo-cardiac IMP levels by multiple bead arrays and QPCR; identification of quantitative trait loci associated with IMPs (ipQTLs) and cdaQTLs; Myocardial IMP levels are associated with CDA.

Concurrent with mouse models, the authors compared genetic variants with CDA in three groups of patients, including 71 anthracycline-treated patients (pediatric cohort); 420 breast cancer patients; a third cohort - cardiac magnetic resonance (CMR) was performed in 24,281 cancer patients. As results\; ipQTLs integrated in genetic models with cdaQTLs accounted for more CDA phenotypic variation than that explained by cda-QTLs alone, including AKT1, MAPK14, MAPK8, STAT3, CAS3 and TP53, are genetic determinants of CDA in patients.

The manuscript is written in standart English, just some re-cheking is requered. The introduction, materials and methods, results, and discussion are well presented. The figures and tables is in a standard, tipical for article processing. The statistical processing of the data is at a very high level. Notes: font unification is required; most of the literature used is not from the half-year 5; please update where possible. The conclusion part is not enouth- need corrections. What are the study limitations?

 Minor editing of English language required

Author Response

Responses de Referee-1

We would like to express our gratitude for the comments made by the referee, which we believe have served to enhance the quality of the manuscript.

Referee_1

The authors propose that innovative research into the levels of intermediate molecular phenotypes (IMPs) in the myocardium associated with histopathological damage may explain sensitivity to CDA anthracyclines, and genes encoding IMPs to identify patients susceptible to this complication. To this purpose, a cohort of mice (n = 165) treated with doxorubicin and docetaxel were studied, quantified: cardiac fibrosis using an Ariol slide scanner and intramyo-cardiac IMP levels by multiple bead arrays and QPCR; identification of quantitative trait loci associated with IMPs (ipQTLs) and cdaQTLs; Myocardial IMP levels are associated with CDA.

Concurrent with mouse models, the authors compared genetic variants with CDA in three groups of patients, including 71 anthracycline-treated patients (pediatric cohort); 420 breast cancer patients; a third cohort - cardiac magnetic resonance (CMR) was performed in 24,281 cancer patients. As results\; ipQTLs integrated in genetic models with cdaQTLs accounted for more CDA phenotypic variation than that explained by cda-QTLs alone, including AKT1, MAPK14, MAPK8, STAT3, CAS3 and TP53, are genetic determinants of CDA in patients.

Point-1: The manuscript is written in standard English, just some re-checking is required.

Response-1: We appreciate the Referee´s constructive feedback. In response, we have thoroughly revised the manuscript to improve the English language usage. All modifications are indicated throughout the document for ease of identification.

The introduction, materials and methods, results, and discussion are well presented. The figures and tables is in a standard, typical for article processing. The statistical processing of the data is at a very high level.

Point-2: Notes: font unification is required;

Response-2: Thank you for your valuable comment. We have revisited the font and unified it.

Point-3:…most of the literature used is not from the half-year 5; please update where possible.

Response-3: We concur with the Reviewer that some of the references cited are indeed dated. However, it is also true that many of these are seminal works in the field that warrant citation, such as the 1978 cardiac biopsy studies in patients with anthracycline-induced cardiomyopathy (Friedman et al., JAMA 1978; Billingham et al., Cancer Treat Rep 1978). Nevertheless, we also agree with the Reviewer that we should cite some of the more influential papers in the field that have been published in the last five years. To this end, we have included eight such references in the revised version of the manuscript, specifically the following:

-Cardinale D, et al., Cardiotoxicity of Anthracyclines. Front Cardiovasc Med. 2020 Mar 18;7:26. doi: 10.3389/fcvm.2020.00026. 

-Kaboré EG, et al., Association of body mass index and cardiotoxicity related to anthracyclines and trastuzumab in early breast cancer: French CANTO cohort study. PLoS Med. 2019 Dec 23;16(12):e1002989. doi: 10.1371/journal.pmed.1002989. 

-Huang J, et al., Understanding Anthracycline Cardiotoxicity From Mitochondrial Aspect. Front Pharmacol. 2022 Feb 8;13:811406. doi: 10.3389/fphar.2022.811406. 

-Mata Caballero R, et al., Incidence of long-term cardiotoxicity and evolution of the systolic function in patients with breast cancer treated with anthracyclines. Cardiol J. 2022;29(2):228-234. doi: 10.5603/CJ.a2020.0062. 

-McOwan TN, et al., Evaluating anthracycline cardiotoxicity associated single nucleotide polymorphisms in a paediatric cohort with early onset cardiomyopathy. Cardiooncology. 2020 May 21; 6:5. doi: 10.1186/s40959-020-00060-0. 

-Wu BB, et al., Mitochondrial-Targeted Therapy for Doxorubicin-Induced Cardiotoxicity. Int J Mol Sci. 2022 Feb 9;23(3):1912. doi: 10.3390/ijms23031912. 

-Zeiss CJ, et al., Doxorubicin-Induced Cardiotoxicity in Collaborative Cross (CC) Mice Recapitulates Individual Cardiotoxicity in Humans. G3 (Bethesda). 2019 Aug 8;9(8):2637-2646. doi: 10.1534/g3.119.400232. 

-Zito C, et al., A multicentre study from the Working Group on Drug Cardiotoxicity and Cardioprotection, Italian Society of Cardiology (SIC). Eur Heart J Cardiovasc Imaging. 2021 Mar 22;22(4):406-415. 

Point-4: The conclusion part is not enough- need corrections. What are the study limitations?

Response-4: Thank you for highlighting this aspect. We have meticulously revised the conclusion section of our paper and have also included a discussion on the limitations of the study to provide a more balanced perspective.

Comments on the Quality of English Language

Point-5: Minor editing of English language required

Response-5: We appreciate your attention to detail. In response to your suggestion, we have conducted a thorough review and revision of the English language usage throughout the manuscript to ensure clarity and precision.

Reviewer 2 Report

The authors have conducted an impressive amount of work and complex analysis, investigating the cardiotoxicity caused by anthracyclines (CDA) and its potential markers to explain susceptibility to CDA. They generated a genetically heterogeneous cohort of mice, induced breast cancer, and treated them with doxorubicin or a combination of doxorubicin and docetaxel. They performed histopathological analysis to quantify heart fibrosis and cardiomyocyte area. They used multiplex bead arrays and qPCR to measure protein and mRNA expression to identify a genetic predisposition to CDA.

Major comments:

 1) I think it is worthy to show how fibrosis changes in untreated animals with age and if it depends on the strain used. Adding data (to Fig2) showing cardiomyocyte area and fibrosis for untreated animals would allow readers to compare the fibrosis area of treated animals with untreated animals and see the level of changes due to the treatment.

2) I do not think the obtained results are well presented and would like to suggest improving the clarity. The text is overloaded with abstract terms and abbreviations, and the main figures are challenging to read, with captions that are not very helpful (e.g., Fig. 3B-I). This makes it difficult to comprehend the work as a whole.

 3) Could the authors:

a) List the gene mutations and changes in protein (or mRNA, miRNA) expression associated with the risk of myocardium damage (high fibrosis) as genetic determinants of CDA in mice and patients.

b) List the intermediate molecular phenotypes associated with fibrosis (myocardium damage), which may not be associated with the risk of CDA. Please state the type of changes observed, such as an increase or decrease in phosphorylation level, protein expression, or mRNA expression. Also, compare these changes with the control group.

4) The authors state (line 431) that "old mice with low levels of H2AX(pS139) in the myocardium had higher global fibrosis in the heart." Phosphorylation of H2AX (a marker of DNA damage) increases due to inflammation and fibrosis. Could you explain why cardiac fibrosis results in an increase in gamma H2AX?

Minor comments:

5) Line 268: "Sen and cols[29]." Did you mean colleagues instead of cols?

6) Line 229: Human genetic analysis

7) Line 277: Can you specify which anthracyclines were given?

8) Line 301: "was evaluated in the four patient cohorts." I believe the authors only mentioned three cohorts: Paediatric Cohort, Breast Cancer, and CMR.

9) Line 331: What does DSV stand for?

10) Line 415: "We quantified the levels of several molecules." (can you specify the molecules?

Figures:

11) Please make all figures more easily readable. The text is often too small, narrow, and lacks sufficient contrast.

12) In Fig.2, please report the actual value or indicate that p>0.05. It is generally accepted to define any abbreviations (including n.s.) that appear in the figure body.

13) In Fig.2A-B, please clarify that the shown p-values are for the comparison of FVB vs F1. Additionally, please indicate the age of FVB and F1 mice.

14) In Fig.3, please add a title and axis labels with units for each figure (3B-3I). The red text does not have good contrast, so please ensure clarity in what is being presented.

15) In Fig.3A-I, it would be helpful to describe the changes in more detail in the results section.

16) In Fig.3C, H, I, when referring to "high and low area," did you mean high and low fibrosis?

17) Fig.5-6, the white text on a yellow and red background is barely visible.

Author Response

Responses to Referee-2

We would like to express our gratitude for the comments provided by the reviewer, which we believe have substantially enhanced the quality of the manuscript.

Referee_2

The authors have conducted an impressive amount of work and complex analysis, investigating the cardiotoxicity caused by anthracyclines (CDA) and its potential markers to explain susceptibility to CDA. They generated a genetically heterogeneous cohort of mice, induced breast cancer, and treated them with doxorubicin or a combination of doxorubicin and docetaxel. They performed histopathological analysis to quantify heart fibrosis and cardiomyocyte area. They used multiplex bead arrays and qPCR to measure protein and mRNA expression to identify a genetic predisposition to CDA.

Major comments:

Point-1. I think it is worthy to show how fibrosis changes in untreated animals with age and if it depends on the strain used. Adding data (to Fig2) showing cardiomyocyte area and fibrosis for untreated animals would allow readers to compare the fibrosis area of treated animals with untreated animals and see the level of changes due to the treatment.

Response-1: Regarding the extent of fibrosis observable in the absence of chemotherapy, in Supplementary Figure S2, we now present an image where almost no increase in damage from chemotherapy is observed. We have expanded this to include five animals, and the quantification is shown (Supplementary Figure S2F). This demonstrates that cardiac damage is minimal in the absence of chemotherapy.

Regarding age-related and intra-strain changes in CDA, we have included comparisons between young and old mice in Supplementary Figure S2. The results demonstrate that F1 mice exhibit more frequent significant changes in CDA compared to FVB mice, indicating their higher sensitivity to anthracycline-induced cardiotoxicity (Supplementary Figure S2G-N) (lines 451-453.).

Indeed, the influence of age on anthracycline-induced cardiotoxicity is a well-established concept. Also, the impact of age on cardiac fibrosis secondary to aging is recognized. The last one is indeed a compelling point, but it must be considered that we would be addressing a different question, namely the influence of age on cardiac fibrosis and, by extension, the genetic regions associated with it and the intramyocardial molecular intermediate phenotypes linked to this fibrosis. We would not be examining the effect of age on chemotherapy-induced cardiotoxicity.

It should be noted that the QTL regions implicated in age-related cardiac damage in the absence of chemotherapy and the intermediate phenotypes are likely, at least in part, distinct from those involved in chemotherapy-induced cardiotoxicity. While this is certainly very intriguing, addressing this other question would necessitate the organization of a new untreated backcross cohort, which would require a substantial amount of time (considering that the oldest mice in our study lived just over two years, as they were treated as they develop breast cancer, to more accurately replicate the circumstances that occur in real life with people) and resources similar to those used for the manuscript presented; this would generate a comparable amount of data, which would be beyond the scope of this work. In any case, a new control cohort generated through backcrossing without treatment would not be considered a suitable control group for the treated cohort, especially from a genetic perspective, as each mouse within these two cohorts possesses unique genetic characteristics. The key aspect lies in assessing the CDA differences within the cohort, specifically, in this case, the treated group.

However, indeed, the Referee is correct in asserting that age is a significant factor, particularly in terms of the contribution of the age variable to the observed cardiotoxicity in the context of chemotherapeutic treatment in mice, which are treated as they develop breast cancer. To address this crucial point, we have incorporated multivariate models of multiple linear regression to examine the influence of age on the degree of observed cardiac fibrosis, which are included in the new Supplementary Table S5. So, we indicated (lines 525-528):

“Furthermore, we conducted multivariate analyses (presented in Table S5) to incorporate the degree of age participation in the context of CDA. The results indicate that age correlated positively with cardiac fibrosis and that the effect of age was particularly significant after the combined therapy (Table S5).”

Point-2. I do not think the obtained results are well presented and would like to suggest improving the clarity. The text is overloaded with abstract terms and abbreviations, and the main figures are challenging to read, with captions that are not very helpful (e.g., Fig. 3B-I). This makes it difficult to comprehend the work as a whole.

Response-2: We have redone the legends of all the main figures and the supplementary ones, we hope to improve their understanding with this. Also, as the Referee indicates later, we have expanded the text of the panels as much as possible and improved the color contrast where required. Not only in Fig.3B-I, but also in all the others when needed. We have included the new version of the figures in the revised manuscript.

Point-3. Could the authors:

Point-3a. List the gene mutations and changes in protein (or mRNA, miRNA) expression associated with the risk of myocardium damage (high fibrosis) as genetic determinants of CDA in mice and patients.

Response-3a: The list of molecular intermediate phenotypes associated with CDA in mice is in Supplementary Table S3.  To make these data work more visually, we have also collected them in a new Supplementary Figure 3A. In this supplementary figure, in the form of a heatmap, we represent the association between a given molecular intermediate phenotype and the CDA with a dot, and the degree of the same is proportional to the area and intensity of the point. Regarding gene mutations, we did not evaluate any gene mutations in mice in this manuscript.

In patients, we have not evaluated levels of proteins, mRNAs, or miRNAs. What we have done is assess whether there are allelic variants in humans (of those same genes that in mice encode molecular intermediate phenotypes associated with CDA) that are associated with CDA in patients. Those participating in human CDA risk are forming the LASSO regression models (Figure 7).

Point-3b.  List the intermediate molecular phenotypes associated with fibrosis (myocardium damage), which may not be associated with the risk of CDA. Please state the type of changes observed, such as an increase or decrease in phosphorylation level, protein expression, or mRNA expression. Also, compare these changes with the control group.

Response-3b: Regarding the list of molecular intermediate phenotypes associated with fibrosis that are not associated with CDA. This statement is not understood since, in mice, myocardial fibrosis is synonymous with CDA. Therefore, those molecules associated with more fibrosis have been associated with more CDA.

As for the comparison with a control group, it would not take place, as we have previously answered in point-1 indicated by the Referee. As we have indicated above, It must be considered that even in the case in which a cohort without treatment had been generated by backcrossing, it would not be properly a control cohort of the treated one, at least from the genetic point of view, because in each of the two cohorts, each mouse would be genetically unique.

Point-4.  The authors state (line 431) that "old mice with low levels of H2AX(pS139) in the myocardium had higher global fibrosis in the heart." Phosphorylation of H2AX (a marker of DNA damage) increases due to inflammation and fibrosis. Could you explain why cardiac fibrosis results in an increase in gamma H2AX?

Response-4: Doxorubicin is a well-known anticancer drug, which is an anthracycline antibiotic. It functions by intercalating into DNA and inhibiting topoisomerase II, an enzyme involved in DNA replication and transcription. This mechanism leads to DNA double-strand breaks, and the cell responds to these breaks by phosphorylating the histone variant H2AX, commonly denoted as γ-H2AX.

Persistent DNA damage, marked by sustained γ-H2AX levels, can lead to cellular senescence or programmed cell death (apoptosis). In the context of the heart, repeated or high-dose exposure to doxorubicin may lead to the accumulation of DNA damage and consequent apoptosis in cardiomyocytes. Since these cells have limited regenerative ability, their loss triggers fibrotic remodeling as a response to maintaining structural integrity, ultimately leading to heart fibrosis.

Here are two papers that provide evidence for this process:

-Vejpongsa, P., & Yeh, E. T. (2014). Prevention of anthracycline-induced cardiotoxicity: challenges and opportunities. Journal of the American College of Cardiology, 64(9), 938-945. This paper discusses the molecular mechanisms of doxorubicin-induced cardiotoxicity.

-Sridharan, V., Tripathi, P., Sharma, S., Corry, P. M., Moros, E. G., Singh, A., ... & Boerma, M. (2019). Cardiac inflammation after local irradiation is influenced by the kallikrein-kinin system. Cancer research, 79(13), 3465-3474. While primarily focused on irradiation, this paper discusses the phosphorylation of H2AX and its link to cardiac damage.

Minor comments:

Point-5.  Line 268: "Sen and cols[29]." Did you mean colleagues instead of cols?

Response-5: Yes, thank you. We have replaced "cols" with "colleagues" (line 324).

Point-6.  Line 229: Human genetic analysis

Response-6: We do not know what the Referee is referring to; on line 229 of the received manuscript (before the changes introduced now), I think there is nothing referring to Human genetic analysis.

Point-7.  Line 277: Can you specify which anthracyclines were given?

Response-7: Within the pediatric cohort, all patients underwent treatment utilizing either doxorubicin, daunorubicin, or epirubicin, conforming to their chemotherapy protocol (Ruiz-Pinto et al., Genomics 2017). In contrast, for the cohort of women diagnosed with breast cancer, epirubicin was the therapeutic agent of choice (Vulsteke et al., Breast Cancer Res Treat. 2015). The cohort evaluated by CMR, comprised breast cancer patients treated with epirubicin, whereas hematological patients received doxorubicin. This pertinent data is incorporated into the methods section in the manuscript's revised edition (lines 338-343).

Point-8.  Line 301: "was evaluated in the four patient cohorts." I believe the authors only mentioned three cohorts: Paediatric Cohort, Breast Cancer, and CMR.

Response-8: Indeed, it is a mistake; there are three cohorts. The error is corrected (line 361).

Point-9.  Line 331: What does DSV stand for?

Response-9: It refers to DNA Sequence Variants, but we have replaced it with SNV (single nucleotide variant), which is the term used in the rest of the manuscript.

.

Point-10.   Line 415: "We quantified the levels of several molecules." (can you specify the molecules?

Response-10: Although the list of evaluated molecules was collected in the supplementary methods, to facilitate their visualization, we moved that section to the main methods. We will rephrase that sentence pointed out by the Referee and cite the relevant methodology section.

Here that, where it said:

We quantified the levels of several molecules in the myocardium after chemotherapy and evaluated their association with heart fibrosis and the cardiomyocyte area (Figure 3A)

Now it says (lines 492-495 with control changes):

We quantified the levels of a number of molecules in the myocardium following chemotherapy (as detailed in the methods section) and assessed their association with cardiac fibrosis and the area of the cardiomyocytes (Figure 3A).

Figures:

Point-11. Please make all figures more easily readable. The text is often too small, narrow, and lacks sufficient contrast.

Response-11: In the new version of the manuscript, we have increased the text size where it has been possible, and we have increased the color contrast.

Point-12. In Fig.2, please report the actual value or indicate that p>0.05. It is generally accepted to define any abbreviations (including n.s.) that appear in the figure body.

Response-12: Thank you for the comment. We have replaced n.s. with the P-value in panels Fig. 2 J and K, which are the only ones that did not have it.

Point-13. In Fig.2A-B, please clarify that the shown p-values are for the comparison of FVB vs F1. Additionally, please indicate the age of FVB and F1 mice.

Response-13: Thank you for the comment. We have added in the legend of Figure 2 that we refer to FVB and F1. As for the age of the FVB and F1 mice, we have to indicate that this is also heterogeneous because they were treated when the animals developed breast cancer, so the age of the mice varies between 51 and 114 weeks of age. We indicate it in the methods section (lines 248-250 in the version with the control changes).

Point-14.  In Fig.3, please add a title and axis labels with units for each figure (3B-3I). The red text does not have good contrast, so please ensure clarity in what is being presented.

Response-14: The axes already have names, such as Dlm1 and Dlm2 (Dlm" stands for Discriminant Linear Method in the context of statistical data analysis). Dlm1" and "Dlm2" in a biplot typically represent the first and second dimensions of a data set reduced by methods such as Principal Component Analysis (PCA) as the one used here."Dlm1" usually corresponds to the first dimension or axis (often the x-axis), which explains the largest amount of variance in the data (indicated as percentage), and "Dlm2" corresponds to the second dimension or axis (often the y-axis), which explains the second largest amount of variance. Each point in the biplot represents an observation (here, a mouse heart with a level of CDA), and the distances between the points are intended to represent the dissimilarities between the observations.

-We have increased the contrast of the red color and the size of the letters.

Point-15. In Fig.3A-I, it would be helpful to describe the changes in more detail in the results section.

Response-15: We have increased the text of the results section referred to this Figure 3, (lines 406-512 with control changes). Where we indicate the following:

"The molecular intermediate phenotypes that correlated with the CDA in the different conditions, collected in Table S3 (single or combined treatment, subepicardial or subendocardial area), were integrated into a principal component analysis and biplot representation. In this way, it was observed how these variables constituted by the molecular intermediate phenotypes helped to differentiate the groups of mice that had greater or lesser cardiotoxicity, in terms of greater or lesser degree of fibrosis (Fig. 3B, D, E-G) and cardiomyocyte area (Fig. 3H, I)."

Point-16. In Fig.3C, H, I, when referring to "high and low area," did you mean high and low fibrosis?

Response-16: No, here we are referring to the area of the cardiomyocytes, which is the other criterion of cardiotoxicity used throughout the manuscript. We mean large or small cardiomyocyte area; we have changed the text of the figure. The legend of Figure 3, like the others, has been rephrased for a better understanding of this.

Point-17. Fig.5-6, the white text on a yellow and red background is barely visible.

Response-17: We have increased the contrast of the lettering in Figure 5, panels D, E, and in Figure 6, panels B-D. We have also increased the size of the lettering as much as we could.

Round 2

Reviewer 1 Report

Тhe authors have taken into account the remarks made. The submitted manuscript is of high originality, quality and of great scientific value.

Reviewer 2 Report

Thank you for the changes.